# A cowpea mosaic virus adjuvant conjugated to liposomes loaded with tumor cell lysates as an ovarian cancer vaccine

Zhongchao Zhao [1,2,3], Debbie K. Ledezma[1,2,3], Jessica Fernanda Affonso de Oliveira[1,2,3], Anthony O. Omole [1,2,3] & Nicole F. Steinmetz [1,2,3,4,5,6,7,8] ✉

Current treatment options for ovarian cancer are limited to surgery to remove tumor tissues and chemotherapy. Although such treatments could provide a short period of remission, most patients still experience recurrent metastatic diseases. Here we present a nanotechnology-based personalized cancer vaccine that can be administrated to patients during the remission stage to prevent recurrent diseases. Autologous tumor cell lysates (TCL) are intriguing, personalized antigens that could be extracted from surgically recovered tumor tissues from patients containing all neoantigens. As proof of concept, we use TCL isolated from a murine ovarian cancer cell line. TCL are first encapsulated in liposomes (TCL-Lip), which are then attached to cowpea mosaic virus (CPMV), a plant virus as a potent adjuvant. Using the ID8-Defb29/Vegf-a-Luc tumor model in female mice, the TCL-Lip-CPMV conjugate vaccine protects mice from tumor challenge by improving antigen processing and presentation, priming an adaptive anti-tumor immunity. Using ovalbumin (OVA) as a model antigen, OVA-Lip-CPMV vaccination protects mice from lung metastasis post-surgical removal of the primary B16F10-OVA dermal tumors. This research establishes a platform by combining two nanoparticle technologies into a single formulation for the simultaneous delivery of antigens and adjuvants, advancing the development of cancer vaccines and immunotherapies.

Cancer immunotherapy has become the fourth pillar of cancer treatments since the approval of immune checkpoint blockade drugs targeting CLTA4, PD1 and PDL1[1–3]. Unlike traditional cancer treatments, immunotherapy helps patients restore their natural immunity cycle and prevents recurrence by establishing antigen-specific anti-tumor immunity[1–3]. However, this approach has yet to succeed with ovarian cancer[4,5] due to the complex immunosuppressive tumor microenvironment (TME)[6]. Advanced-stage ovarian cancer is one of the deadliest gynecological cancers in females[7,8], with 19,710 new cases and 13,270 deaths predicted for 2023 in the USA[9]. Standard treatments

[1]Aiiso Yufeng Li Family Department of Chemical and Nano Engineering, University of California, San Diego, 9500 Gilman Dr., La Jolla, CA 92093, USA. [2]Center for Nano-ImmunoEngineering, University of California, San Diego, 9500 Gilman Dr, La Jolla, CA 92093, USA. [3]Moores Cancer Center, University of California, San Diego, 9500 Gilman Dr., La Jolla, CA 92093, USA. [4]Department of Bioengineering, University of California, San Diego, 9500 Gilman Dr., La Jolla, CA 92093, USA. [5]Department of Radiology, University of California, San Diego, 9500 Gilman Dr., La Jolla, CA 92093, USA. [6]Institute for Materials Discovery and Design, University of California, San Diego, 9500 Gilman Dr., La Jolla, CA 92093, USA. [7]Center for Engineering in Cancer, University of California, San Diego, 9500 Gilman Dr., La Jolla, CA 92093, USA. [8]Shu and K.C. Chien and Peter Farrell Collaboratory, University of California, San Diego, 9500 Gilman Dr., La Jolla, CA 92093, USA. ✉e-mail: nsteinmetz@ucsd.edu

involve surgical resection to remove the primary malignant tumor tissues followed by chemotherapy. Although such procedures could benefit patients with a short period of remission, up to 70% of patients experience recurrence, which significantly reduces the survival rate and quality of life[10]. Surgery also leads to postoperative trauma and the establishment of a new TME primed for recurrence and metastasis[11–13]. Therefore, the remission stage post-surgery provides an opportunity to introduce additional treatment strategies to prevent recurrence and prolong survival. A personalized cancer vaccine that could be administrated shortly after to patients post their surgical operation and within the short period of remission would achieve this goal by effectively reprograming the TME, restoring the immunity cycle, and establishing tumor-specific anti-tumor immunity.

Effective cancer vaccines generally require two major components: tumor antigens and adjuvants. Many tumor-associated antigens (TAAs) have been identified, such as EGFR for lung and breast cancer[14,15], HER2 for ovarian and breast cancer[15,16], and NY-ESO1 for testicular cancer[17]. Some tumor-specific antigens (TSAs) are also known, such as human papillomavirus (HPV) E6/E7 for HPV[+] cancer[18,19]. However, malignant cells can often escape these narrowly-targeted immune responses due to resistance mechanisms and tumor heterogeneity. Tumor cell lysates (TCL) provide a superior source of antigens because they contain all tumorigenic proteins, broadening the anti-tumor immune response and making it less likely that tumor cells will escape[20]. Most ovarian cancer patients undergo surgery to remove tumor tissues, providing an abundant source of autologous TCL containing patient-specific and tumor-specific antigens[21]. TCL have therefore been explored for the development of cancer vaccines and immunotherapies[22–28]. A potent adjuvant is required to promote antigen uptake and processing by immune cells, helping to establish tumor antigen-specific immunity. Such immunostimulatory adjuvants include CpG oligodeoxynucleotides[29] and poly-IC[30], which are Toll-like receptor (TLR) agonists, and STING agonists[31,32]. Recently, TCL were packaged in liposomes with the TLR9 agonist CpG to develop an immunotherapy for triple-negative breast cancer, improving the therapeutic efficacy in a tumor mouse model[22].

We previously reported that cowpea mosaic virus (CPMV), a plant virus, functions as a potent immunostimulatory adjuvant for cancer immunotherapy. CPMV has an RNA genome encapsulated in a ~30 nm icosahedral capsid[33]. The capsid proteins trigger TLR2 and TLR4 while the encapsulated viral RNAs trigger TLR7[34,35]. CPMV does not infect or replicate in mammalian cells[36], but it is recognized as a pathogen-associated molecular pattern (PAMP)[37–45] and can therefore stimulate the immune system. As a triple-TLR agonist, CPMV has exceptional adjuvant potency and can stimulate the innate immune system to reprogram the suppressive TME[33–35,46–48]. CPMV's anti-tumor efficacy has been confirmed in murine models of melanoma, breast cancer, colorectal cancer, glioma, and ovarian cancer – importantly efficacy was also confirmed by treatment of companion dogs with oral melanoma and mammary tumors[33,47–52]. From an engineering perspective, CPMV forms stable nanoparticles with surface-exposed lysine residues, facilitating chemical engineering for combinational immunotherapy approaches[53–56].

In this work, we conjugate TCL-loaded liposomes (TCL-Lip) to our CPMV particles to develop an antigen–adjuvant combination immunotherapy (TCL-Lip-CPMV) for ovarian cancer. As proof of concept, we use a murine metastatic ovarian cancer model, ID8-Defb29/Vegf-a-Luc, in female C57BL/6J mice[57]. ID8-Defb29/Vegf-a-Luc is derived based on ID8 cell line, which has 213 identified nonsynonymous mutations including common antigens such as LNPEP, NDUDS6, MYO15, and CDK15[58]. We note that in prior work[59], we established a vaccine by conjugating irradiated cancer cells to CPMV; while efficacy was observed in a mouse model, a stringent vaccination schedule had to be established and likely only select antigens were processed in this formulation. To improve efficacy and broaden the antigens for a more

robust anti-tumor immune response, we isolate TCL from cultured ID8-Defb29/Vegf-a-Luc cells and load it into liposomes—as a model of autologous TCL. We first examine the co-delivery of TCL and CPMV to antigen-presenting cells (APCs) and the immunomodulatory functions. The efficacy of the TCL-Lip–CPMV vaccines is then evaluated in mice with ID8-Defb29/Vegf-a-Luc ovarian tumors using a prophylactic setup. We further assay efficacy in a B16F10-OVA model, which allow us to establish efficacy post-surgery to prevent recurrence and outgrowth of metastasis. Using the B16F10-OVA model also provides a means to study ovalbumin as a model antigen to delineate immune mechanisms.

## Results and discussion
### Production of TCL and TCL-laden liposomes
TCL were derived from the syngeneic murine ovarian cancer cell line ID8-Defb29/Vegf-a-Luc, which resembles human high-grade serous carcinoma (HGSC)[57]. Confluent cells were harvested, washed, and exposed to five freeze–thaw cycles, and the supernatant containing the TCL was recovered by centrifugation[22] (Supplementary Fig. 1a). The TCL were then loaded into 1,2-dioleoyl-*sn*-glycero-3-phosphocholine (DOPC; 90% molar) and 1,2-dipalmitoyl-*sn*-glycero-3-phosphoethanolamine-*N*-azido(polyethylene glycol)–2000 (DPPE-PEG2K-azide; 10% molar) liposomes by thin-film rehydration[22,60]. The heterogeneous liposome population was processed by extrusion to generate more uniform 100-nm liposomes, as confirmed by dynamic light scattering (DLS) (Supplementary Fig. 1b). Unpackaged TCL was removed by tangential flow filtration (TFF) until no proteins were detected in the flow-through (FT) fraction by NuPAGE and UV–Vis spectrophotometry (Supplementary Fig. 1c, d). The average TCL loading capacity was $103.4 \pm 13.4\,\mu g$ TCL/mg lipids based on six independent preparations (Supplementary Fig. 1e). The liposome formulation was imaged by cryo-electron microscopy (cryo-EM) (Supplementary Fig. 1f).

### Conjugation of TCL-laden liposomes to CPMV
After loading TCL into liposomes, we conjugated CPMV to the liposomes to form adjuvanted conjugate (hereafter TCL-Lip–CPMV, Fig. 1a) using azide and DBCO click chemistry (Fig. 1b). Azide groups were introduced during liposome formulation using the DPPE-PEG2K-azide lipids. The surface-exposed lysine residues of CPMV were then functionalized with an alkyne handle using NHS chemistry[33,61,62]. CPMV was therefore conjugated to DBCO-PEG$_4$-NHS ester, a bifunctional linker, to produce CPMV-DBCO[63]. Purified CPMV-DBCO particles were similar to native CPMV in terms of absorbance (A260/280 ~ 1.7) indicating the presence of intact particles containing RNA[64] (Supplementary Fig. 2a). DLS revealed the CPMV-DBCO particles were ~30 nm in diameter (Supplementary Fig. 2b) and transmission electron microscopy (TEM) confirmed that CPMV and CPMV-DBCO were intact and monodisperse (Supplementary Fig. 2c). The CPMV-DBCO particles were then mixed with the DPPE-PEG2K-azide-functionalized TCL-Lip to allow tethering via click reaction. The success of the reaction was first confirmed by the additional higher molecular weight bands on NuPAGE (Fig. 1c). The large (L) and small (S) coat proteins of CPMV have molecular weights of ~42 and ~24 kDa[65]. In the TCL-Lip–CPMV sample, higher molecular weight bands corresponding to L-PEG2K-DPPE (~44.7 kDa) and S-PEG2K-DPPE (~26.7 kDa) in addition to the unconjugated L and S proteins were detected (Fig. 1c); the difference in molecular weight corresponds to the molecular weight of the lipids, confirming that DBCO modified CPMV was conjugated to the DPPE-PEG2K-azide lipids on liposomes. Successful tethering of CPMV to the TCL-Lip was further confirmed by size exclusion chromatography (SEC), showing that TCL-Lip–CPMV eluted as a single peak rather than two different peaks corresponding to the elution of liposomes and CPMV alone (Fig. 1d). No free proteins were observed at end of the column volume as shown by the full elution profiles (Supplementary Fig. 3), indicating that conjugation to CPMV did not damage the liposomes. Cryo-EM images showed that most of the liposomes were

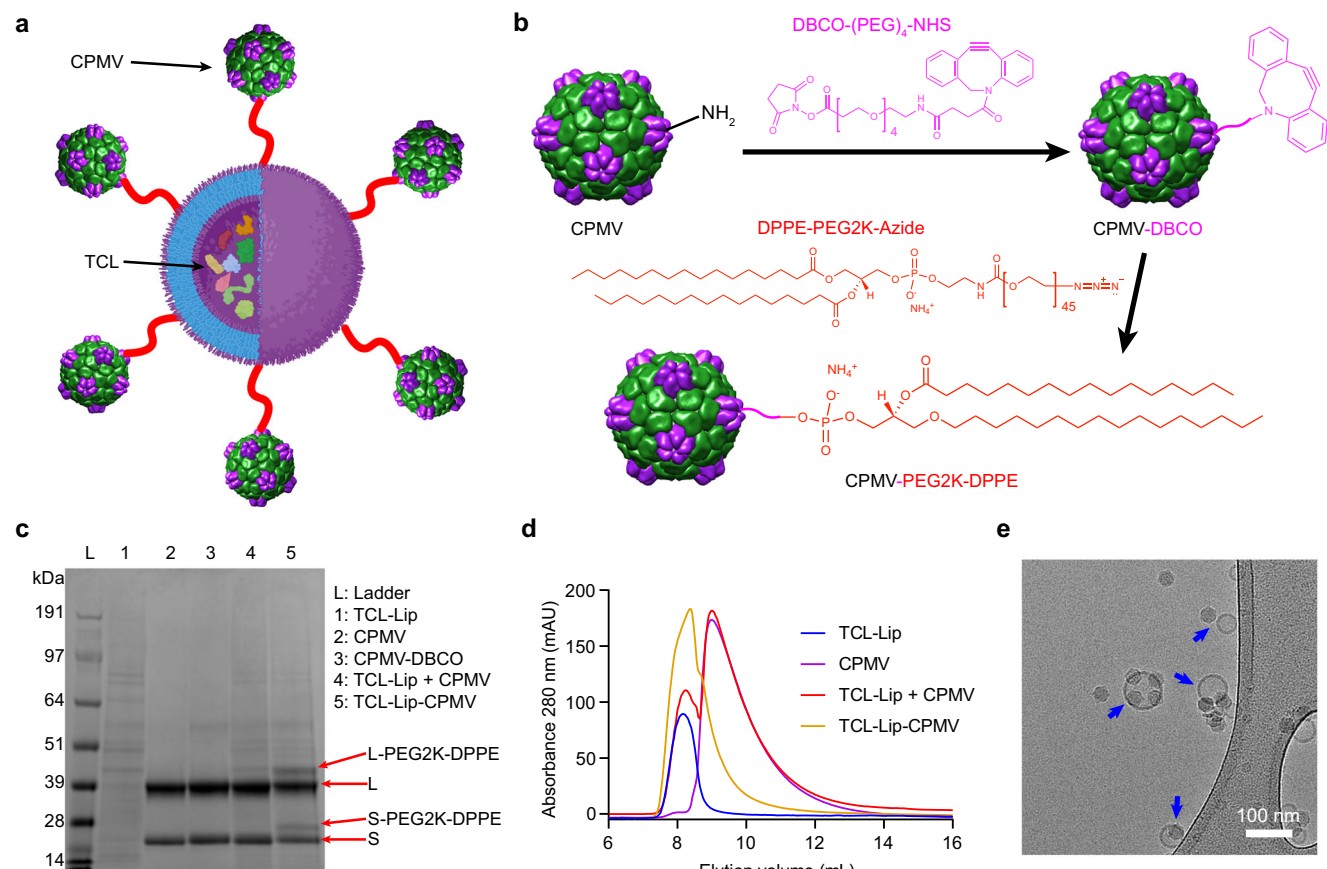

**Fig. 1 | Conjugation of cowpea mosaic virus (CPMV) to tumor cell lysates (TCL)-loaded liposomes (TCL-Lip) using click chemistry. a** Structure of the TCL-Lip–CPMV complex. **b** The two-step conjugation process: CPMV is functionalized with the bifunctional linker (DBCO-PEG₄-NHS ester) via NHS chemistry to produce CPMV-DBCO, which is then reacted with azide groups on the DPPE-PEG2K-azide liposomes to produce TCL-Lip–CPMV. **c** NuPAGE of TCL-loaded liposome, native CPMV, CPMV-DBCO, the TCL-Lip + CPMV mixture, and the TCL-Lip–CPMV complex. The complex features two CPMV subunit proteins (large (L) and small (S)) and

two additional bands (L-PEG2K-DPPE and S-PEG2K-DPPE). **d** SEC analysis of TCL-Lip (blue), CPMV (violet), TCL-Lip + CPMV (red), and TCL-Lip−CPMV (orange). The complex elutes as a single peak at the same column volume as the liposomes, whereas the TCL-Lip + CPMV mixture elutes as two peaks corresponding to the separate components. **e** Cryo-EM image of TCL-Lip−CPMV (blue arrows) showing CPMV closely associated with liposomes. Three independent experiments were performed with similar results (**c**–**e**). Part of **a** is generated by BioRender. Source data are provided as a Source Data file (**c**).

closely associated/tethered with CPMV particles (Fig. 1e) and different numbers of CPMV-DBCO were conjugated per liposome. In stark contrast, the CPMV and liposome nanoparticles did not show any signs of association in simple mixtures (Supplementary Fig. 4). In the TCL-Lip−CPMV formulation, we also observed free CPMV-DBCO particles separated from liposomes, indicating that conjugation was not 100% and could be further optimized. We further performed DLS analysis of the TCL-Lip−CPMV (Supplementary Fig. 5); although no obvious size increase was observed, we confirmed that the conjugation of CPMV to liposome did not generate aggregations with a distribution of the nanoparticles around 100 nm in size similar to TCL-Lip alone (Supplementary Fig. 1b).

## Co-delivery of TCL and CPMV in vitro and in vivo

We determined whether TCL-Lip−CPMV enables the co-delivery of TCL and CPMV particles to APCs. We first labeled TCL with Oregon Green 488 (OG488) and CPMV with Cy5. OG488-TCL was produced by incubating purified TCL with OG488 maleimide and OG488 carboxylic acid succinimidyl ester to label the free thiol and amine groups; successful protein labeling was confirmed by NuPAGE (Supplementary Fig. 6a). We produced CPMV-DBCO-Cy5 and CPMV-Cy5 particles by incubating CPMV with sulfo-Cy5-NHS ester with or without DBCO-PEG₄-NHS ester. NuPAGE and agarose gel electrophoresis confirmed

the conjugation of Cy5 (Supplementary Fig. 6b and c). The UV−Vis spectra of CPMV, CPMV-Cy5, and CPMV-DBCO-Cy5 particles indicated a similar degree of labeling with ~55 Cy5 molecules per CPMV-Cy5 and ~44 per CPMV-DBCO-Cy5 (Supplementary Fig. 6d). DLS confirmed that CPMV-Cy5 and CPMV-DBCO-Cy5 were intact measuring ~30 nm in diameter (PDI: 0.03), the same as native CPMV (Supplementary Fig. 6f) and both particle types were found to be intact and monodisperse by TEM (Supplementary Fig. 6f). After unconjugated free dye molecules were removed, OG488-TCL were loaded into 100 nm liposomes as described above. We then prepared OG488-TCL-Lip + CPMV-Cy5 mixtures and conjugated the two components by mixing OG488-TCL-Lip and CPMV-DBCO-Cy5. Agarose gel electrophoresis showed separate signals for CPMV and TCL-Lip in all lanes except OG488-TCL-Lip−CPMV-Cy5, where the OG488 and Cy5 signals were colocalized−therefore indicating successful conjugation of the two nanoparticle systems (Fig. 2a). NuPAGE showed two additional bands in the OG488-TCL-Lip-CPMV-Cy5 lane, further confirming the presence of CPMV coat proteins linked with the lipids (Fig. 2b).

Having formulated the different fluorescent constructs, we first compared their uptake and co-delivery in vitro using bone marrow-derived dendritic cells (BMDCs) isolated and prepared from C57BL/6J mice[55]. OG488-TCL-Lip-CPMV-Cy5 and formulations of the individual and mixed components were incubated with BMDCs for 1 and 24 h.

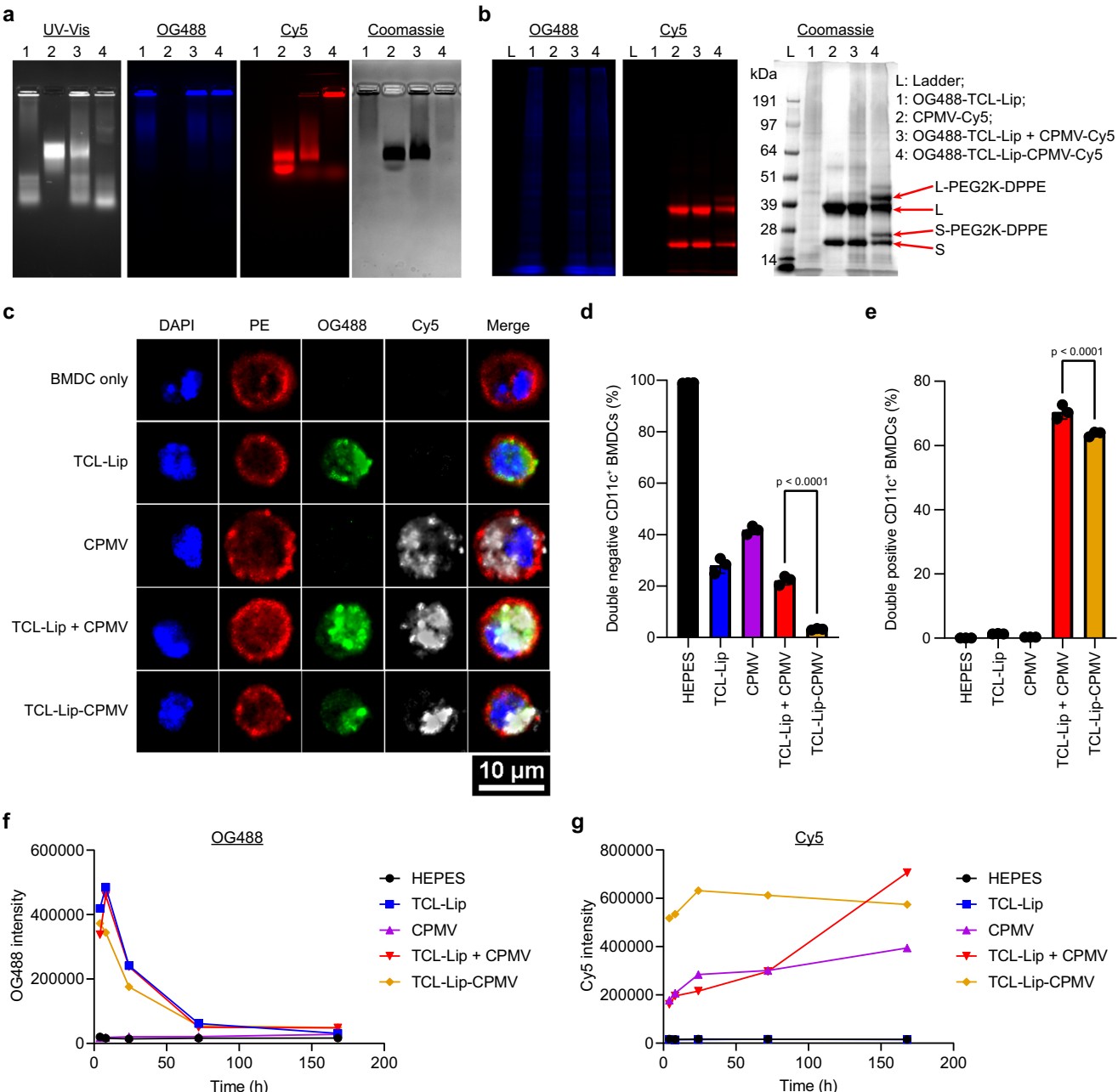

**Fig. 2 | Co-delivery of labeled TCL-Lip–CPMV in vitro and in vivo. a** Analysis of Oregon Green 488 (OG488)-TCL-Lip, CPMV-Cy5, OG488-TCL-Lip + CPMV-Cy5, and OG488-TCL-Lip-CPMV-Cy5 by agarose gel electrophoresis. Gel was imaged for RNA (UV–Vis, white), TCL (OG488, blue), CPMV (Cy5, red), and protein (Coomassie). **b** Analysis of the same constructs by NuPAGE. **c** Confocal images of bone marrow-derived dendritic cells (BMDCs) 24 h post-incubation with OG488-TCL-Lip, CPMV-Cy5, OG488-TCL-Lip + CPMV-Cy5, and OG488-TCL-Lip-CPMV-Cy5, showing the co-delivery of OG488-TCL-Lip and CPMV-Cy5. Nuclei (DAPI, bule), membranes (PE, red), TCL (OG488, green), CPMV (Cy5, red). **d, e** Flow cytometry data for double-negative BMDCs (**d**) and double-positive BMDCs (**e**) 24 h post-incubation with HEPES control, OG488-TCL-Lip, CPMV-Cy5, OG488-TCL-Lip + CPMV-Cy5, and OG488-TCL-Lip-CPMV-Cy5. $n = 3$ independent experiments; data are expressed as mean ± SD. **f, g** Fluorescence intensity of OG488 and Cy5 in draining popliteal lymph nodes over time; n = 2 independent experiments from 4 to 72 h. Three independent experiments were performed with similar results (**a**–**c**). Color schemes in (**d**–**g**): HEPES control (black), TCL-Lip (blue), CPMV (violet), TCL-Lip + CPMV (red), TCL-Lip–CPMV (orange). Statistical significance was determined by ordinary one-way ANOVA. Source data are provided as a Source Data file (**a, b, d**–**g**).

TCL-Lip and/or CPMV were taken up by BMDCs (Fig. 2c and Supplementary Fig. 7a). Differences between OG488-TCL-Lip-CPMV-Cy5 and controls were observed by flow cytometry. After 1 h incubation, ~40% of BMDCs were double-negative (no uptake of TCL-Lip or CPMV) in the TCL-Lip, TCL-Lip + CPMV, and TCL-Lip–CPMV groups and >70% were double-negative in the CPMV group–the reduced uptake of CPMV vs. the liposomes is likely because BMDCs prefer to take up larger particles (Supplementary Fig 7b). Importantly, after 24 h, few of the BMDCs were double-negative in the OG488-TCL-Lip-CPMV-Cy5 group whereas >20% of BMDCs remained double-negative without either or both components (Fig. 2d). While there were more double-positive BMDCs in the TCL-Lip + CPMV group (26%) than the TCL-Lip–CPMV group (16%) after 1 h (Supplementary Fig. 7c), after 24 h the difference was marginal, with 64% double-positive for TCL-Lip–CPMV and 70% for TCL-Lip + CPMV (Fig. 2e). Overall, this study confirmed that liposomes and CPMV are efficiently taken up by BMDC and conjugating CPMV to

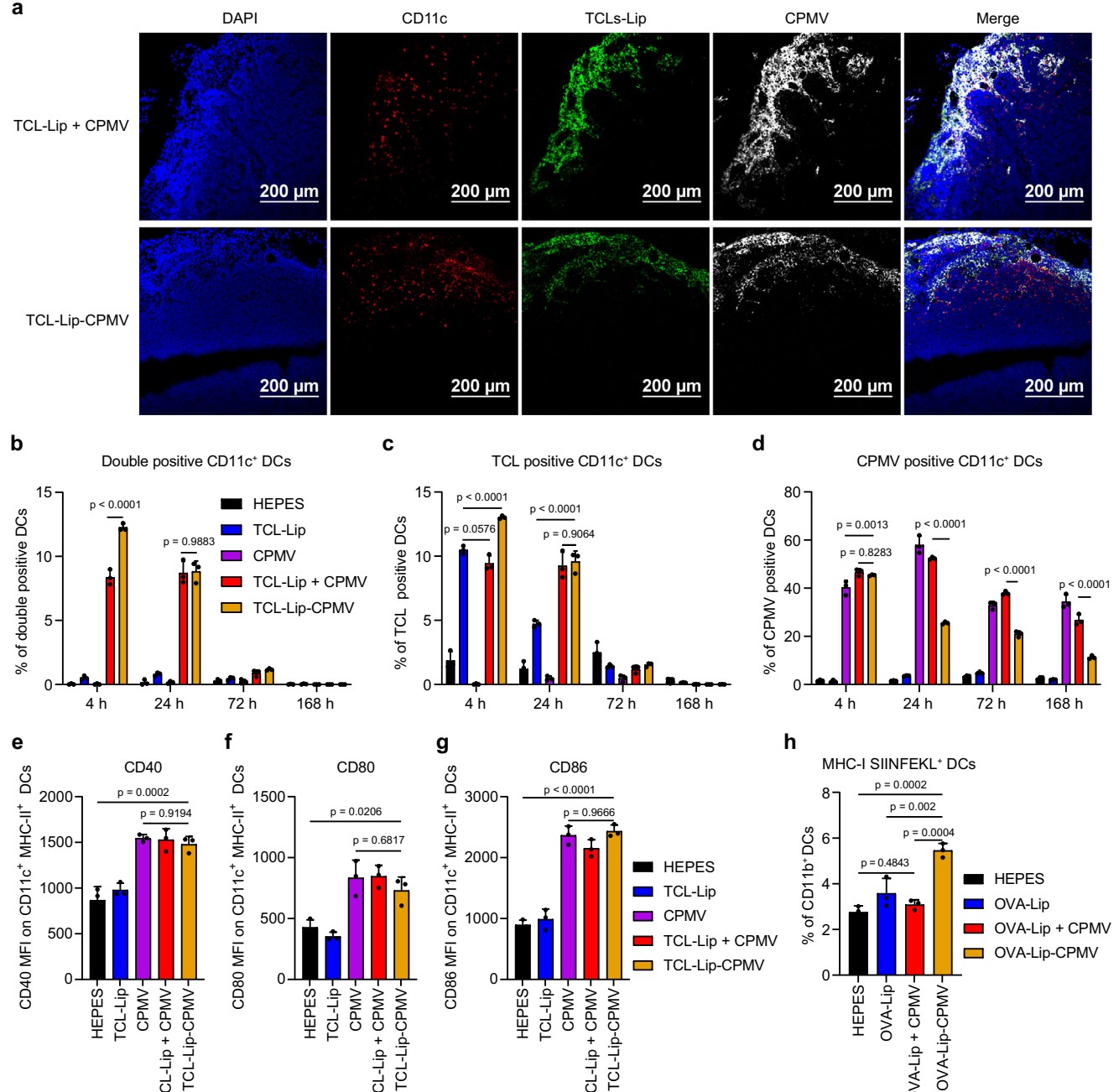

**Fig. 3 | Quantification of co-delivered TCL and CPMV in popliteal draining lymph nodes and activation of popliteal lymph nodes. a** Confocal images of harvested lymph node sections 4 h after footpad injections of TCL-Lip + CPMV and TCL-Lip−CPMV. Dendritic cells (DCs) are stained with a PE-conjugated anti-CD11c antibody. Nuclei (DAPI, blue), membranes (PE, red), TCL (OG488, green), CPMV (Cy5, red). **b**−**d** Flow cytometry analysis of TCL⁺/CPMV⁺ (**b**), TCL⁺ (**c**), CPMV⁺ (**d**) CD11c⁺ DCs in harvested lymph nodes based on OG488 and Cy5 fluorescence, $n = 3$ independent experiments; data are expressed as mean ± SD. **e**−**g** Activation of DCs in popliteal draining lymph nodes 24 h after footpad injections. The activation status of CD11c⁺ MHC-II⁺ DCs in lymph nodes was determined by flow cytometry using expression levels of CD40 (**e**), CD80 (**f**), and CD86 (**g**) surface markers; $n = 3$ independent experiments; data are expressed as mean ± SD. **h** SIINFEKL presentation by MHC-I on CD11c⁺ DCs 48 h post footpad injections of different vaccine formulations; $n = 3$ independent experiments; data are expressed as mean ± SD. Color schemes in (**b**−**h**): HEPES control (black), TCL-Lip or OVA-Lip (blue), CPMV (violet), TCL-Lip + CPMV or OVA-Lip + CPMV (red), TCL-Lip−CPMV or OVA-Lip-CPMV (orange). Statistical significance was determined by ordinary one-way ANOVA. Source data are provided as a Source Data file (**b**−**h**).

liposomes does not interfere with the uptake of the formulated nanoparticles by BMDCs in vitro.

Next, we determined whether the TCL-Lip−CPMV formulation could improve the co-delivery of TCL and CPMV in vivo. The labeled TCL-Lip−CPMV and other formulations were injected subcutaneously (s.c.) into the footpads of C57BL/6 J mice (Supplementary Fig. 8a and b). Popliteal draining lymph nodes (PDLNs) were harvested 4, 8, 24, 72, and 168 h post-injection for IVIS imaging and confocal microscopy. IVIS imaging revealed the presence of TCL and CPMV in the PDLNs (Supplementary Fig. 8c). OG488 fluorescence from TCL-Lip was the most intense after 4 and 8 h followed by rapid intensity decay within 72 h, which indicated that TCL was quickly processed within 3 days inside the PDLNs (Fig. 3f). In contrast, Cy5 fluorescence from CPMV remained relatively stable over the time course of the experiment for all groups,

indicating that CPMV was slowly processed in PDLNs compared to TCL (Fig. 3g). Cy5 fluorescence in the TCL-Lip−CPMV group was stronger than the other CPMV-Cy5 groups, indicating more CPMV was taken up by lymphocytes. Confocal images of lymph node sections also indicated the colocalization of OG488 and Cy5 with DCs, macrophages, B cells, and T cells in the TCL-Lip + CPMV and TCL-Lip−CPMV groups after 4 and 24 h (Fig. 3a, Supplementary Figs. 9–16). More importantly, even though most of the TCL-Lip and CPMV signals were colocalized in the TCL-Lip + CPMV group, the CPMV signal had penetrated deeper into the lymph nodes, whereas almost all the TCL-Lip and CPMV were colocalized in the TCL-Lip−CPMV group. These results indicate that conjugation led to synchronous trafficking of TCL-Lip and CPMV together, whereas the components were trafficked independently when presented as a mixture, resulting in the deeper penetration of CPMV.

To examine the uptake of our formulations by CD11c[+] DCs and F4/80[+] macrophages in the lymph nodes, single-cell suspensions were prepared from the PDLNs and analyzed by flow cytometry. Significantly more TCL and CPMV particles were co-delivered to DCs and macrophages in the TCL-Lip−CPMV group vs the mixture after 4 h, but the difference became negligible after 72 h (Fig. 3b and Supplementary Fig. 17a). Similarly, the number of DCs and macrophages positive only for TCL at the 4 h time point was higher in the TCL-Lip−CPMV group, but this was not the case for CPMV delivery (Fig. 3c and Supplementary Fig. 17b). Although conjugating CPMV to liposomes did not improve the delivery of CPMV to DCs and macrophages at the early time point, it dramatically changed the CPMV residence time possibly due to the faster digestion of CPMV as demonstrated by the faster drop in the number of CPMV[+] DCs and macrophages over time compared to TCL-Lip + CPMV and CPMV groups (Fig. 3d and Supplementary Fig. 17c). Additionally, throughout the time course of the experiment, the TCL-Lip + CPMV and CPMV groups showed similar percentages of CPMV[+] DCs and macrophages, confirming that CPMV was trafficked freely and independently when mixed with liposomes. Overall, these data show that conjugating CPMV to TCL-Lip could improve the co-delivery of both components to APCs, especially at the early time point (4 h post-injection).

## Activation of DCs within draining lymph nodes

To determine whether TCL-Lip−CPMV complexes can activate APCs, we administered the unlabeled formulations by s.c. injection into the footpads and characterized the activation status of CD11c[+] MHC-II[+] DCs after 4 and 24 h by measuring the levels of surface markers CD40, CD80, and CD86. After 4 h, CD40 and CD80 remained at basal levels in all groups and CD86 remained at basal levels in the TCL-Lip group and PBS control but was elevated -1.3 fold in all three groups containing CPMV (Supplementary Fig. 18). After 24 h, CD40, CD80 and CD86 were elevated -1.6 fold (CPMV), -2 fold (TCL-Lip + CPMV), and -2.5 fold (TCL-Lip−CPMV, respectively, relative to the TCL-Lip group and HEPES control (Fig. 3e–g). There was no significant difference between the TCL-Lip group and HEPES control, ruling out immune stimulation by DOPC, DPPE-PEG2K-azide lipids, or the TCL. However, there was also no significant difference between the CPMV, TCL-Lip + CPMV, and TCL-Lip−CPMV groups, suggesting that CPMV could stimulate the DCs effectively and the addition of the liposome has a negligible effect on APC stimulation. This result agrees with our findings that CPMV itself is an effective immunostimulatory adjuvant[55] for immune cell stimulation and liposomes have no immunostimulatory functions.

Next, we investigated whether the liposome and CPMV conjugation can achieve tumor antigen presentation on DCs within the draining lymph nodes. Because TCL contains many different tumor proteins, we replaced TCL with a single protein—ovalbumin (OVA)—as a model antigen to identify the presented MHC-I peptide SIINFEKL on DCs. First, we generated 100 nm OVA-laden liposomes (OVA-Lip) (Supplementary Fig. 19a) and used the click chemistry approach described above to produce OVA-Lip−CPMV complexes (Supplementary Fig. 19b and c) analogous to TCL-Lip. All formulations were then s.c. injected into the footpad; 48 h later, PDLNs were harvested and processed to analyze the presentation of MHC-I SIINFEKL on CD11c[+] DCs. OVA-Lip−CPMV exhibited significantly more MHC-I SIINFEKL-positive CD11c[+] DCs compared to all other groups (Fig. 3h), demonstrating the advantage of conjugating CPMV to liposomes for antigen−adjuvant co-delivery and antigen presentation within APCs.

## Modulation of the adaptive immune system within the i.p. space and spleen

To investigate whether the TCL-Lip−CPMV vaccine candidate can modulate the immune cell population within the intraperitoneally (i.p.) space and spleen, we i.p. injected the different formulations into healthy female C57BL/6J mice weekly for three weeks ($n = 5$ per group); 7 days after the third injection, mice were sacrificed (day 18 post-inoculation, Supplementary Fig. 20a), and cells isolated from the i.p. washes and spleens were dissociated into splenocyte suspensions for analysis by flow cytometry (Supplementary Fig. 20b). As observed in our previous study[48], more CD4[+] and CD8[+] T cells infiltrated into the i.p. space for all the treatment groups involving CPMV compared to the TCL-Lip group and HEPES control (Fig. 4a and f), resulting from the immunomodulatory nature of CPMV[48].

With regard to the distribution of CD4[+] T cells in the i.p. space, CPMV, CPMV + TCL-Lip, and TCL-Lip−CPMV groups showed similar percentages of memory (CD44[+]) T cells (Fig. 4b) including the effector memory (CD44[+]CD62L[−]) T cells, higher than the TCL-Lip group and HEPES control (Fig. 4c). In contrast, the CPMV, CPMV + TCL-Lip and TCL-Lip−CPMV groups featured significantly less central memory (CD44[+]CD62L[+]) T cells compared to the TCL-Lip and HEPES controls (Fig. 4d). In our previous study, we discovered that CPMV can reside within the i.p. space for over 7 days[66]. Therefore, this result may be attributed to the prolonged CPMV residing within the i.p. space, therefore increasing the duration of antigen exposure, which favors CD62L[Lo] memory CD4 T cells[67]. Interestingly, the TCL-Lip−CPMV group featured significantly more activated (CD69[+]) CD4 T cells than all other groups. With regard to the distribution of CD8[+] T cells, there was no significant difference between the TCL-Lip −CPMV and TCL-Lip + CPMV groups, but there was an increasing trend in the overall number of CD8[+] T cells and the proportions of CD44[+] memory T cells, CD44[+]CD62L[−] effector memory T cells, CD44[+]CD62L[+] central memory T cells, and CD69[+] activated T cells in the following sequence: HEPES → TCL-Lip → CPMV → TCL-Lip + CPMV → TCL-Lip−CPMV (Fig. 4f–j). However, the analysis of splenocytes by flow cytometry showed no differences among the T-cell populations in all five groups (Supplementary Fig. 21). Overall, all vaccine formulations containing CPMV improved the infiltration of CD4[+] and CD8[+] T cells into the i.p. space, whereas the addition of TCL-Lip, especially when conjugated to CPMV, further augmented CD8[+] T-cell infiltration and activation in the i.p. space. It is of note though that the improvement was not statistically significant when comparing the TCL-Lip−CPMV to CPMV alone and TCL-Lip + CPMV. We reason that although CPMV alone is an effective immune adjuvant for immune stimulation, the further improved infiltration and activation of CD8[+] T cells in the i.p. space of the TCL-Lip−CPMV vaccination could potentially lead to better protection against tumor challenge.

Despite no major statistical difference being observed for the overall CD8[+] T cells from the i.p. space and spleen of the TCL-Lip−CPMV, we did observe improved OVA antigen presentation on DCs when conjugating CPMV to OVA-Lip (Fig. 3h). Therefore, we hypothesized that the liposome and CPMV conjugation might lead to more infiltration and expansion of antigen-specific CD8[+] T cells. To test this, we produced OVA-Lip−CPMV (Supplementary Fig. 19) and probed whether SIINFEKL antigen-specific CD8[+] T cells are elicited. We first

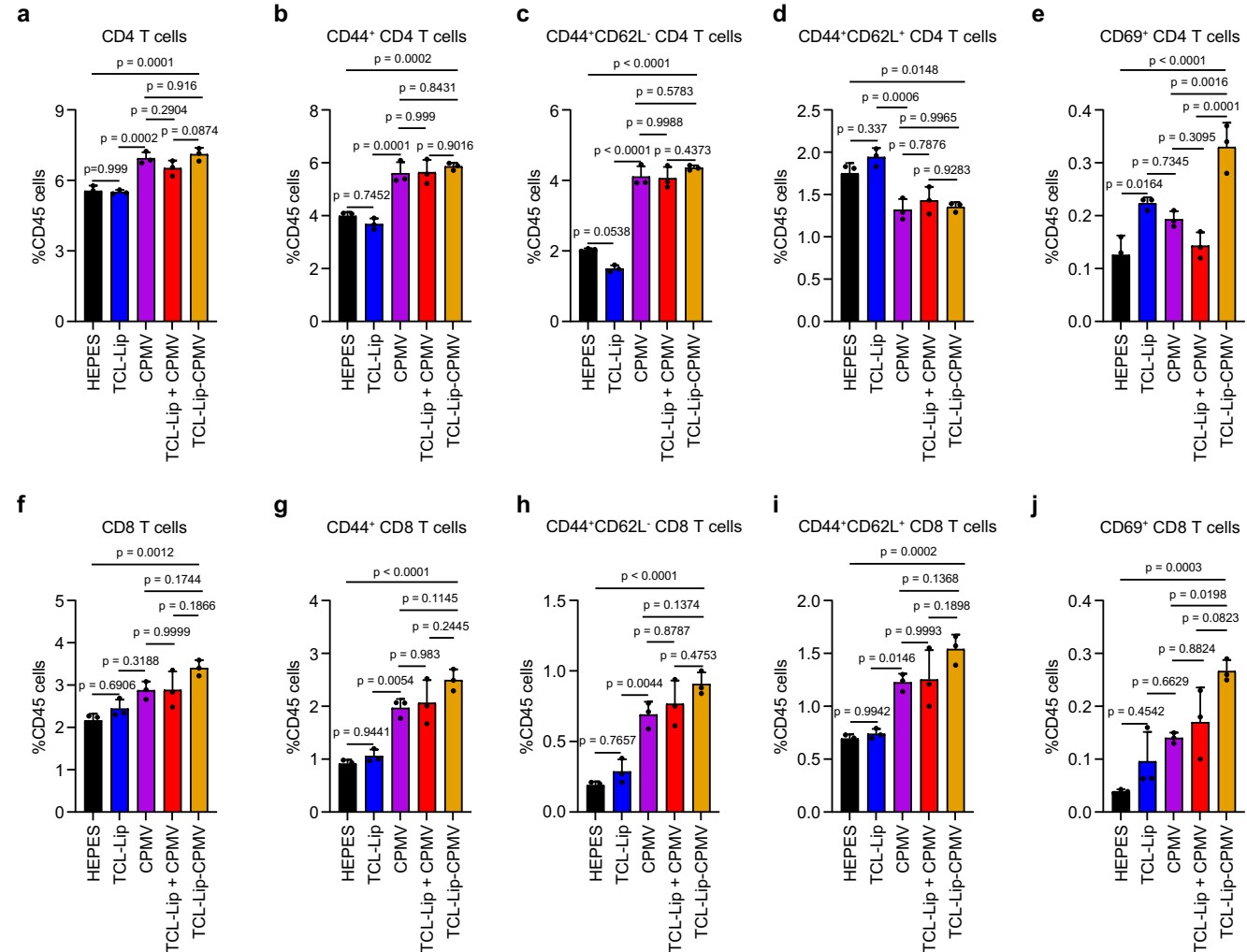

**Fig. 4 | CD4⁺ and CD8⁺ T cells within the i.p. space after treatment. a** Percentage of CD3⁺CD4⁺ T cells among CD45⁺ immune cells. **b** Percentage of CD44⁺ memory CD4⁺ T cells among CD45⁺ immune cells. **c** Percentage of CD44⁺CD62L⁻ effector memory CD4⁺ T cells among CD45⁺ immune cells. **d** Percentage of CD44⁺CD62L⁺ central memory CD4⁺ T cells among CD45⁺ immune cells. **e** Percentage of CD69⁺ activated CD4⁺ T cells among CD45⁺ immune cells. **f** Percentage of CD3⁺CD8⁺ T cells among CD45⁺ immune cells. **g** Percentage of CD44⁺ memory CD8⁺ T cells among CD45⁺ immune cells. **h** Percentage of CD44⁺CD62L⁻ effector memory CD8⁺ T cells

among CD45⁺ immune cells. **i** Percentage of CD44⁺CD62L⁺ central memory CD8⁺ T cells among CD45⁺ immune cells. **j** Percentage of CD69⁺ activated CD8⁺ T cells among CD45⁺ immune cells. $n = 3$ independent experiments; data are expressed as mean ± SD. Color schemes in (**a**–**j**): HEPES control (black), TCL-Lip (blue), CPMV (violet), TCL-Lip + CPMV (red), TCL-Lip–CPMV (orange). Statistical significance was determined by ordinary one-way ANOVA. Source data are provided as a Source Data file.

vaccinated C57BL/6 J mice ($n = 5$) using two i.p. doses of OVA-Lip, CPMV, OVA-Lip + CPMV, and OVA-Lip−CPMV at weekly intervals in a prime-boost vaccination regimen. Seven days after the second dose, i.p. washes and spleens were harvested and processed to identify SIINFEKL⁺ CD8⁺ cells in an MHC-I dextramer assay (Supplementary Fig. 22a). We observed significantly more SIINFEKL⁺ CD8⁺ T cells in the OVA-Lip−CPMV group compared to all other groups in both the i.p. washes and splenocytes (Fig. 5 and Supplementary Fig. 22b). In the i.p. space, the proportion of SIINFEKL⁺ cells among all CD8⁺ cells were ~5.4% for the OVA-Lip−CPMV group compared to ~4.2% for OVA-Lip + CPMV, ~2.8% for CPMV, and ~2% for OVA-Lip (Fig. 5a). In the spleen, the proportion of SIINFEKL⁺ CD8⁺ cells remained below 1.5% in all groups but was highest in the OVA-Lip−CPMV group (~1.3%) and below 1% in the others (Fig. 5b and c). Therefore, we hypothesize that although conjugating CPMV to TCL-Lip failed to significantly improve the overall CD8⁺ T cells in the i.p. space and spleen post-vaccination, it could generate more antigen-specific CD8⁺ T cells (this was shown using OVA as a testbed) that can recognize tumor antigens, and therefore protect mice against tumor challenge.

## TCL-Lip−CPMV protects mice against ovarian cancer challenge

First, we investigated whether the TCL-Lip−CPMV vaccine can protect mice from ovarian cancer using the ID8-Defb29/Vegf-a-Luc ovarian cancer model in C57BL/6J mice. The i.p. ID8-Defb29/Vegf-a-Luc tumors grow aggressively with ascites and disseminated tumor nodules on the peritoneal membranes and intestines—while this mimics human disease, it makes this model a challenging model for surgery. Therefore, instead of surgery to mimic the remission phase, we carried out the experiments in a prophylactic setting, i.e. administering the nano-vaccine prior to tumor challenge. We first administered the first dose of vaccine candidates (and controls) on day −3 in healthy mice, which was treated as the remission stage post-surgery. On day 0, all mice were then challenged with cancer cells ($5 \times 10^6$ ID8-Defb29/Vegf-a-Luc cells i.p.) to mimic the cancer recurrence. Starting from day 4, each mouse received five further i.p. doses of TCL-Lip−CPMV and other controls at weekly intervals. Each treatment comprised 30 μg TCL within liposomes and 100 μg CPMV. We used unconjugated CPMV and TCL-Lip (individually or mixed) as controls, as well as a HEPES buffer control, with $n = 8$ mice per treatment group (Fig. 6a). Tumor

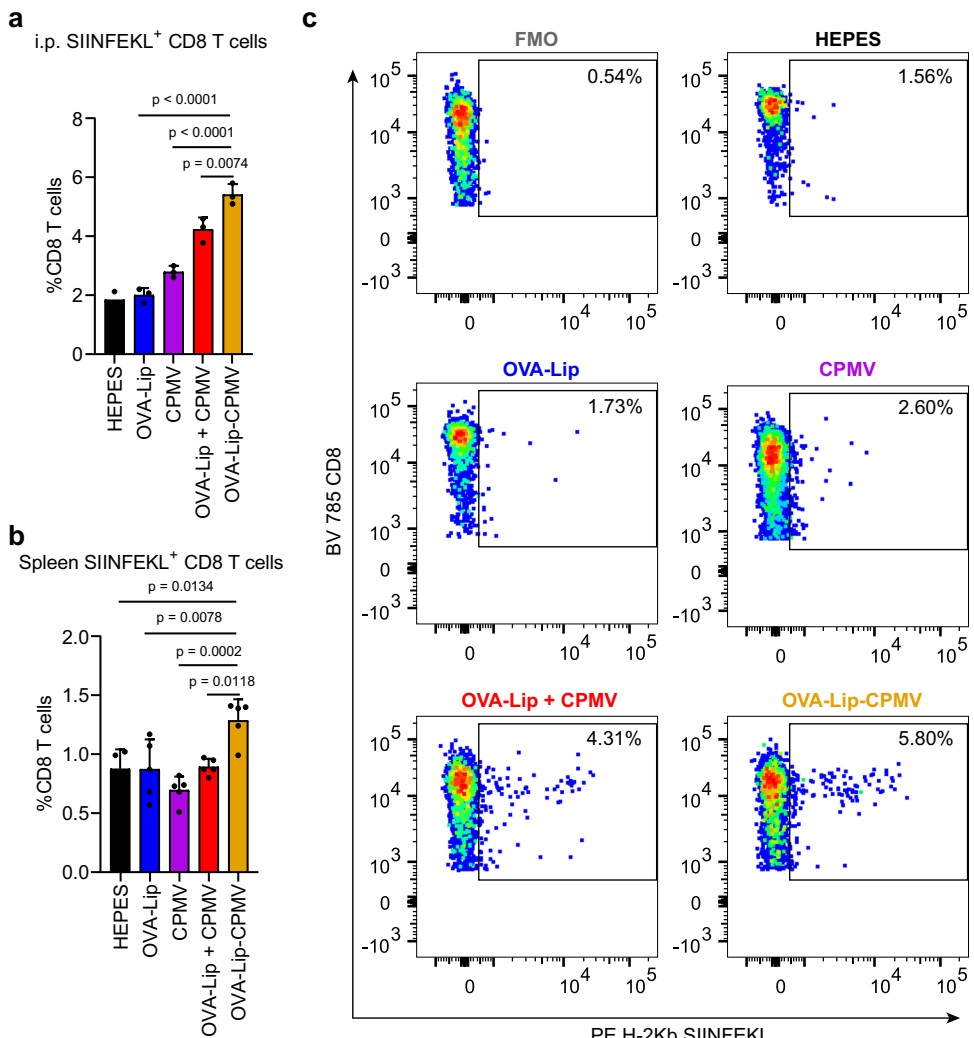

**Fig. 5 | MHC-I specific CD8+ T cells. a, b** SIINFEKL+ CD8+ T cells within the i.p. space (**a**) and spleens (**b**) were quantified by flow cytometry using a PE-conjugated MHC-I SIINFEKL dextramer. $n = 2$ independent experiments for HEPES control and $n = 3$ independent experiments for other groups in (**a**) and $n = 4$ independent experiments for HEPES and $n = 5$ independent experiments for other groups in (**b**); data are expressed as mean ± SD for all groups with a sample size n ≥ 3. **c** Representative flow cytometry plots of SIINFEKL+ CD8+ T cells within the i.p. space of different treatment groups. Color schemes in (**a, b**): HEPES control (black), TCL-Lip (blue), CPMV (violet), TCL-Lip + CPMV (red), TCL-Lip−CPMV (orange). Statistical significance was determined by ordinary one-way ANOVA. Source data are provided as a Source Data file (**a, b**).

progression was monitored by measuring body weight gain (Supplementary Fig. 23) and increases in body circumference (Fig. 6c) resulting from the tumor burden and ascites. Mice were euthanized if their body weight exceeded 35 g or their circumference exceeded 9 cm[33]. Following our previously established protocol for this tumor model, mice that were tumor-free on day 100 were treated as survivors.

Overall, the TCL-Lip−CPMV treatments led to the best protection against ovarian cancer challenge in terms of survival rate, with 5 out 8 mice remained tumor-free followed by the CPMV group (2/8, $p < 0.247$) and TCL-Lip + CPMV mixture group (1/8, $p < 0.064$) (Fig. 6b). TCL-Lip alone had no significant effect, similar to the HEPES control. Although TCL-Lip−CPMV group did not show statistical significance compared to TCL-Lip + CPMV and CPMV for survival, its efficacy was confirmed by its undefined median survival, whereas all other groups had a defined median survival: 45 days for the HEPES group, 42 days for the TCL-Lip group, 64 days for the TCL-Lip + CPMV group, and 69 days for the CPMV group. This result was further corroborated by the individual tumor growth curves: CPMV alone or mixed with the liposomes delayed the induction and growth of ovarian tumors, albeit at lower efficacy compared to the TCL-Lip−CPMV treatment, whereas

liposome alone and HEPES showed no effect in tumor onset and growth (Fig. 6c). This data indicates that CPMV and TCL-Lip + CPMV improve the protection against tumor challenge and median survival for treated mice compared to HEPES and TCL-Lip, but the co-delivery of CPMV and TCL using the TCL-Lip−CPMV formulation increases the efficacy and survival benefit even further. Throughout the course of the study, all tumor-free mice that received TCL-Lip + CPMV and TCL-Lip−CPMV showed no sign of sickness such as weight loss. We also prepared oxidized TCL (OxTCL) and irradiated TCL (IrTCL) and loaded them into liposomes to determine whether chemical oxidation or X-ray exposure of TCL would improve the efficacy even more (Supplementary Figs. 23 and 24). However, the original TCL-Lip−CPMV outperformed all other groups and there was no improvement by using OxTCL or IrTCL (Supplementary Fig. 24b and c).

### OVA-Lip−CPMV protects mice against dermal melanoma
Next, we considered the B16F10-OVA model using C57BL/6J mice—this not only allowed us to analyze antigen processing (Fig. 3h) and antigen-specific immunity (Fig. 5) based on the model antigen OVA but also enabled us to establish a surgery model. First, we formulated

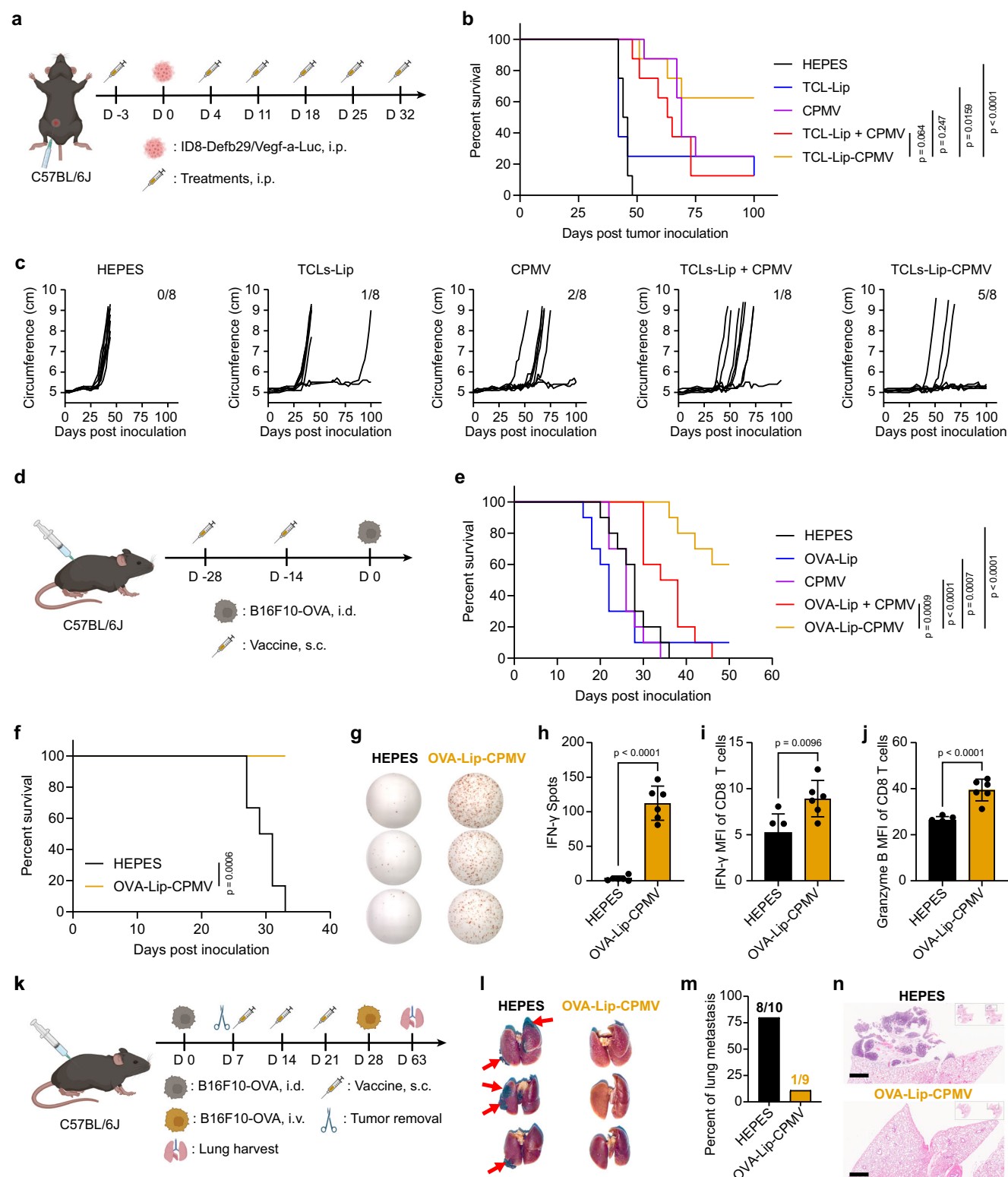

OVA-Lip, OVA-Lip + CPMV, and OVA-Lip–CPMV and tested whether these vaccine candidates would protect mice from tumor challenges. Mice were vaccinated via a prime-boost regimen (bi-weekly apart) containing 30 μg OVA and/or 100 μg CPMV in 100 μL HEPES; CPMV and HEPES served as controls. Then mice were challenged by intradermal (i.d.) injection of B16F10-OVA cells ($2 \times 10^5$ cells in 20 μL PBS) on the right flank (Fig. 6d). Tumor volumes were measured every two days starting on day 7 post-inoculation. All mice were sacrificed when tumor volumes reached 1000 mm³ and mice free of tumors 50 days

post-tumor inoculation were treated as survivors. The OVA-Lip–CPMV vaccination protected mice from tumor challenge, resulting in significant survival benefits compared to all other groups with 6 out of 10 mice remaining tumor-free 50 days post-tumor inoculation. In stark contrast, all mice (but one in the OVA-Lip group) were sacrificed due to tumor burden before the study end-point (50 days) (Fig. 6e and Supplementary Fig. 25a). This data is consistent across the two animal models demonstrating that synchronous delivery of tumor antigens (here the model antigen OVA) and CPMV adjuvant achieve effective

**Fig. 6 | Treatment efficacy of CPMV-conjugated liposomes. a** Treatment and tumor injection schedule for the i.p. ovarian cancer model. The following groups were tested: HEPES, TCL-Lip, CPMV, TCL-Lip + CPMV, and TCL-Lip−CPMV (*n* = 8 per group). Each treatment consisted of 100 μg CPMV and/or 30 μg TCL in TCL-Lip. **b** Survival rates for all treatment groups. **c** Individual circumferences in each group. The number of tumor-free mice 100 days post-inoculation is shown in the top right corner. **d** Vaccination and tumor injection schedule for the i.d. melanoma model (prevention study). The following groups were tested: HEPES, ovalbumin (OVA)-Lip, CPMV, OVA-Lip + CPMV, and OVA-Lip−CPMV (*n* = 10 per group). Each vaccine consisted of 100 μg CPMV and/or 30 μg OVA in OVA-Lip. **e** Survival rates for all groups. **f** Surviving animals were subjected to rechallenge; data show the survival rates for the B16F10-OVA rechallenge study (n = 6 for the Control and OVA-Lip−CPMV groups). **g** Representative images of the IFN-γ ELISpot assay for the splenocytes stimulated by SIINFEKL peptides for the Control and OVA-Lip−CPMV groups. **h** Quantification of the IFN-γ spots for the Control and OVA-Lip−CPMV

groups, *n* = 6 independent experiments; data are expressed as mean ± SD. Flow cytometry analysis of the intracellular IFN-γ (**i**) and Granzyme B (**j**) within CD8 T cells post splenocytes and B16F10-OVA incubation; *n* = 6 independent experiments; data are expressed as mean ± SD. **k** Schedule for i.d. melanoma challenge, tumor removal by surgery, vaccination, and lung metastasis establishment. **l** Images of harvested lungs showing tumor metastasis. **m** Percentage of mice showing lung metastasis in both the control and OVA-Lip−CPMV groups. **n** H&E histology analysis of harvested lungs. Color schemes in (**b, e, f, h–j, m**): HEPES control (black), TCL-Lip or OVA-Lip (blue), CPMV (violet), TCL-Lip + CPMV or OVA-Lip + CPMV (red), TCL-Lip−CPMV or OVA-Lip-CPMV (orange). Statistical significance in **b, e**, and **f** was calculated using the log-rank (Mantel−Cox) test. Statistical significance in (**h–j**) was calculated using the unpaired parametric *T*-test with Two-tailed *p*-value. Parts of **a,d**, and **k** are generated by BioRender. Source data are provided as a Source Data file (**h–j**).

immune protection against tumor onset and growth. To test the robustness of the anti-tumor immunity, mice free of tumors after 50 days post-tumor inoculation were rechallenged i.d. on the left flank using $2 \times 10^5$ B16F10-OVA cells in 20 μL PBS and age-matched mice as controls. All the survivors continued to reject tumor growth, suggesting that the vaccination established long-lasting anti-tumor immunity against B16F10-OVA melanoma (Fig. 6f and Supplementary Fig. 26a–c). Finally, there were no signs of body weight loss resulting from the treatment (Supplementary Fig. 25b).

To correlate efficacy with mechanism, we analyzed T-cell responses using splenocytes from the surviving mice that had received the OVA-Lip−CPMV vaccines (we also analyzed the one survivor from the OVA-Lip group and data are shown in Supplementary Fig. 26f–h). Splenocytes from OVA-Lip−CPMV and untreated control mice were stimulated for 48 h using OVA 257-264 SIINFEKL peptides and live B16F10-OVA cells and then analyzed by ELISpot assay. Splenocytes from the OVA-Lip−CPMV group showed IFN-γ response against SIINFEKL peptide, indicating that OVA-Lip−CPMV vaccination led to the establishment of a pro-inflammatory immune response against the OVA antigen (Fig. 6g, h). More importantly, splenocytes from the OVA-Lip−CPMV group showed overwhelmingly more IFN-γ spots compared to control group (Supplementary Fig. 26e), demonstrating that the splenocytes could recognize tumor antigens on cancer cells. Flow cytometry experiment further confirmed the improved IFN-γ and Granzyme B production within CD8 T cells from the OVA-Lip−CPMV vaccinated mice (Fig. 6i, j). We also stained for intracellular Perforin; however, no significant changes were observed (Supplementary Fig. 26d). Together data support that OVA-Lip−CPMV vaccination established an antigen-specific anti-tumor immunity by recognizing the OVA antigen, therefore protecting from B16F10-OVA tumor challenge. Additional future experiments should assess whether OVA-Lip−CPMV induces antigen expansion to achieve broad antigen recognition and protection in vaccinated mice after rejecting B16F10-OVA challenge. This could be tested by ELISpot assay and flow cytometry after stimulating splenocytes with its parental B16F10 cells and by rechallenging survived mice using B16F10 tumors.

## OVA-Lip−CPMV vaccination post-surgical removal of the primary tumor protects from lung metastasis

Data consistently showed enhanced efficacy of the liposome and CPMV conjugated formulations (TCL-Lip−CPMV and OVA-Lip−CPMV) vs. control groups. To minimize the number of animals, we only focused on the OVA-Lip−CPMV formulation to investigate whether the liposome and CPMV conjugated formulation have the potential in preventing metastatic disease post-surgical removal of the primary tumor—the model was set for vaccine administration during the remission stage. Of note, a shortcoming of this study is that comparisons to control groups such as CPMV only or OVA-Lip + CPMV

cannot be made; here we only focus on the OVA-Lip−CPMV group. First, i.d. B16F10-OVA tumors were established as described above, and then surgically removed 7 days post-tumor inoculation prior to the metastasis. Mice then received three s.c. doses of the OVA-Lip−CPMV vaccine containing 30 μg OVA and 100 μg CPMV on days 7, 14, and 21; HEPES served as control. To resemble the recurrence of metastatic disease, on day 28, animals were challenged intravenously (i.v.) using $1 \times 10^5$ B16F10-OVA cells in 100 μL PBS to establish lung metastasis. On day 63, mice were sacrificed, and lungs were harvested and examined for metastasis (Fig. 6k). OVA-Lip−CPMV vaccine effectively prevented B16F10-OVA lung metastasis with 1 out 9 mice showed tumor nodes on lungs compared to 8 out of 10 mice from the control group (Fig. 6l, m). This result was further confirmed by H&E histology staining (Fig. 6n).

## Toxicity analysis

Lastly, we performed a biodistribution and histology study to gain insights into the safety of the vaccine candidates. We administered a single dose of fluorescently labeled TCL-Lip−CPMV or TCL-Lip + CPMV and collected i.p. washes, serum, and major organs 24 h and 7 days post-treatment to determine possible acute vs. chronic effects. At the 24 h time-point, CPMV and TCL were detected in spleen, liver, kidneys, and heart independent of conjugation status; at 7 days the signals were no longer detectable indicating clearance (Fig. 7A and Supplementary Fig. 27a). CPMV and TCL were not detectable in i.p. washes or serum at 24 h or 7 days post-administration, consistent with rapid clearance from circulation (Fig. 7b and Supplementary Fig. 27b). All organs collected showed normal histology (Fig.7c and Supplementary Fig. 27c), indicating no apparent toxicity.

We have designed an immunotherapeutic modality (TCL-Lip−CPMV) that combines a plant virus adjuvant with TCL-loaded liposomes aiming to prevent the recurrence of (ovarian) cancer. In the clinic, ovarian cancer patients undergo surgical debulking and thus tumor tissue is available to produce autologous TCL as a personalized source of tumor antigens—we provide a potential vaccine formulation strategy. As proof-of-concept, liposomes were loaded with TCL isolated from a murine ovarian cancer cell line and then tethered with a plant virus-based adjuvant (CPMV). It should be noted that a shortcoming of using TCL from a cell line is that this does not realistically mimic autologous TCL from established tumors, because these lysates would contain not only tumor-associated and neoantigens but also normal proteins and immunosuppressive cells and cytokines, which may interfere with vaccine efficacy. Future studies thus must also consider testing with autologous TCL from established tumors which would mimic the clinical scenario more realistically. Because a surgery model could not be established for ovarian cancer (due to the technical difficulty of survival surgery and removal of intraperitoneal tumors), we designed a prophylactic study to mimic the patient's

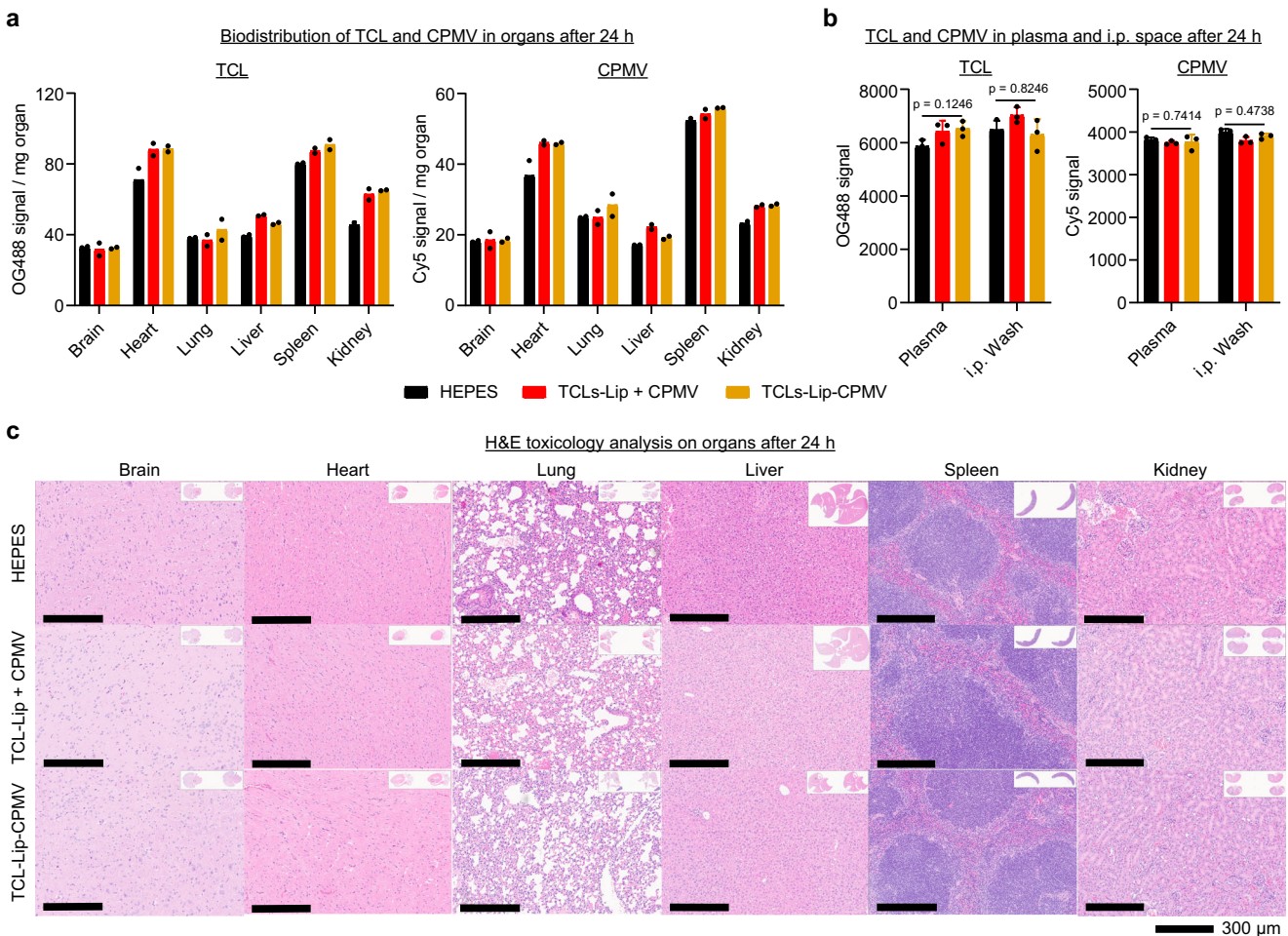

**Fig. 7 | Biosafety of formulated vaccines.** Biodistribution of i.p. injected fluorescently labeled TCL and CPMV from different vaccine groups in organs (**a**) and i.p. space and blood plasma (**b**) at 24 h. $n = 2$ independent experiments for (**a**) and $n = 3$ independent experiments for (**b**); data are expressed as mean ± SD for (**b**). **c** H&E toxicology analysis of major organs at 24 h after vaccination of different formulations. Color schemes in (**a**, **b**): HEPES control (black), TCL-Lip + CPMV (red), TCL-Lip −CPMV (orange). Statistical significance was determined by ordinary one-way ANOVA. Source data are provided as a Source Data file (**a**, **b**).

remission phase and efficacy studies showed that the TCL-Lip−CPMV vaccine candidate demonstrated improved potency over controls (mixed or single components) in terms of improved tumor suppression and survival benefit. Mechanistically, the TCL-Lip−CPMV vaccine achieves the simultaneous delivery of CPMV and TCL to DCs and macrophages in draining lymph nodes; and using OVA as a model antigen, we demonstrated that OVA-Lip−CPMV conjugate leads to enhanced antigen presentation in the draining LNs over controls. Moreover, TCL-Lip−CPMV established adaptive immunity in the i.p. space, promoting T-cell infiltration. Although there was no significant difference observed for the overall CD8[+] T cells when comparing the conjugated TCL-Lip−CPMV formulation to the simple mixing formulation, using OVA to replace TCL, we confirmed that the conjugated formulation indeed improved the production of antigen-specific CD8[+] T cells. Moreover, the OVA-Lip−CPMV demonstrated exceptional efficacy in preventing B16F10-OVA melanoma onset and development of lung metastasis (the latter in a post-surgery model). Mechanism studies confirmed the establishment of antigen-specific anti-tumor immunity, highlighting that synchronous delivery of antigen and adjuvant enabled through the combination of nanoparticle technologies results in enhanced cancer vaccine potency. Overall, this versatile platform allows the loading of various agents such as drugs, antigens, and nucleic acids into liposomes, followed by conjugation with CPMV as an adjuvant. This will facilitate the

development of cancer vaccines and combination therapies with enhanced efficacy.

## Methods

### Ethical statement

All studies involving mice were carried out in accordance with the guidelines of the Institutional Animal Care and Use Committee (IACUC) of the University of California, San Diego (UCSD) under protocol number S18021 and were approved by the Animal Ethics Committee of UCSD.

Since ovarian cancer only afflicts women, we obtained 7-week-old female C57BL/6J mice from Jackson Laboratories for all our experiments in this study. All mice were housed in a light-controlled room with a 12h-light/12h-dark cycle at 20–23 °C and 50% ± 20% humidity, and all mice had free access to water and diet. For the ID8-Defb29/ Vegf-a-Luc tumor model, following the approved protocol, mice were euthanized when their body weight reached 35 g or their circumference reached 9 cm. For the B16F10-OVA model, following the approved protocol mice were euthanized when their tumor volumes reached 1000 mm³. In all mouse models, maximal tumor size/burden was not exceeded.

**Production of CPMV and CPMV conjugates.** CPMV was propagated in No. 5 black-eyed pea plants and purified using previously

established protocols[59,68]. To propagate CPMV, 20 μL of 0.1 mg/mL CPMV in potassium phosphate (KP) buffer (pH 7.0) was mechanically inoculated per leave using 7-day-old No. 5 black-eyed pea plants. Two weeks later, all leaves were harvested and frozen at −80 °C. To extract CPMV, the frozen leaves were blended with three volumes of pre-chilled 0.1 M KP buffer (pH 7.0) at 4 °C, then passed through Miracloth (Fisher Scientific, NC1510798) to collect the CPMV-containing solution, which was then centrifuged at 18,500×g for 20 min at 4 °C to remove plant debris. The resulting supernatant was mixed with 0.7 volumes of a 1:1 chloroform (Fisher Scientific, AA31068M6) to 1-butanol (Fisher Scientific, AA32614K7) solution and incubated for 30 min at 4 °C. The mixture was then centrifuged at 6600×g for 10 min at 4 °C to collect the upper aqueous phase. Sodium chloride (0.2 M; Fisher Scientific, S271-500) and 10% w/v PEG 8000 (Fisher Scientific, BP233-1) were then added into the collected solution and mixed overnight at 4°C. Following an overnight reaction, the mixture was centrifuged at 30,000 × g for 15 min at 4 °C to precipitate all proteins. The resulting pellet was then resuspended in 10 mL pre-chilled 0.1 M KP (pH 7.0) and centrifuged at 13,500 × g for 15 min at 4 °C; CPMV was present in the supernatant. CPMV was further purified using sucrose gradient ultracentrifugation by loading the supernatant onto a 10–40% sucrose gradient. After 2.5 h centrifugation at 28,000 rpm (SW 32 Ti rotor, Beckman Coulter) at 4 °C, the sucrose fractions containing CPMV were collected in a dark room. The collected fractions were centrifuged again at 42,000 rpm (Type 50.2 Ti rotor, Beckman Coulter) for 2.5 h at 4 °C to pellet CPMV, which was then resuspended in PBS (Corning, 21-040-CV), and stored at −20 °C until use.

CPMV has solvent-exposed surface lysine residues, which can be used to attach molecules through N-hydroxysuccinimide (NHS) chemistry. To produce CPMV-DBCO, CPMV-Cy5 and CPMV-DBCO-Cy5, DBCO-PEG₄-NHS ester (649.7 g/mol, BroadPharm, BP-22288) and/or sulfo-Cy5 NHS ester (777.95 g/mol, Lumiprobe, 43320) were simply mixed and incubated with CPMV. In detail, for CPMV-DBCO, 2 mg/mL (final concentration) of CPMV (molecular weight = $5.6 \times 10^6$ g/mol) was mixed with 1200-molar excess of DBCO-PEG₄-NHS ester in 1 mL of PBS and incubated at room temperature for 2 h; for CPMV-Cy5, 2 mg/mL of CPMV was mixed with 1500-molar excess of sulfo-Cy5 NHS ester in 1 mL of PBS; for CPMV-DBCO-Cy5, 2 mg/mL of CPMV was mixed with 1200-molar excess of DBCO-PEG₄-NHS ester and 1500-molar excess of sulfo-Cy5 NHS ester in 1 mL of PBS. Post-incubation, Amicon Ultra-0.5 mL centrifugal filters with a 100-kDa cutoff membrane (Millipore Sigma, UFC510096) were used for purification by centrifugation at 10,000×g for 5 min; following first centrifugation, 0.4 mL of 50 mM HEPES (pH 7.4) (Fisher Scientific, BP310-100) was then loaded into each centrifugal filters to dilute the concentrated reaction solution and centrifuged again. This step was repeated 4 times. After the last centrifugation, concentrated and purified CPMV-DBCO, CPMV-Cy5, and CPMV-DBCO-Cy5 particles were stored at 4 °C.

**Cell culture and production of TCL.** ID8-Defb29/Vegf-a-Luc cell line was engineered to express luciferase based on the ID8-Defb29/Vegf-a-Luc cell line[57,69]. No authentication of this line was performed in this current study. ID8-Defb29/Vegf-a-Luc cells were cultured in RPMI 1640 medium (Corning, 10-040-CV) containing 10% (v/v) fetal bovine serum (FBS) (Fisher Scientific, A5670801), 0.05 mM β-mercaptoethanol (Thermo Fisher Scientific, 21985023), 1 mM sodium pyruvate (Thermo Fisher Scientific, 11360070), and 1% (v/v) penicillin/streptomycin (Pen/Strep) (Cytiva, SV30010). Cells were maintained at 37 °C and 5% CO₂. To harvest cells for intraperitoneal tumor inoculation, cells were first treated with Trypsin–EDTA (Corning, 25-052-CI) for 5 min, then collected and washed by PBS 3 times; after the last washing step, cells were resuspended in PBS at a defined concentration and placed on ice for injections.

To produce tumor cell lysates (TCL), after harvesting cells using trypsin–EDTA, cells were washed 3 times using PBS and then

resuspended in 50 mM HEPES (pH 7.4) at a concentration of $1 \times 10^7$ cells/mL. Cells were then subjected to 5 cycles of freeze and thaw process using liquid nitrogen and 37 °C water bath. To recover soluble TCL and remove cell debris, after the 5th freeze and thaw process, the cell mixture was centrifuged at 14,000×g for 10 min twice. The supernatant TCL was collected and stored at −80 °C for further experiments. The TCL protein concentration was determined using a bicinchoninic acid (BCA) assay kit (Thermo Fisher Scientific, 23225) on a Tecan plate reader.

To track TCL for in vitro and in vivo experiments, TCL was labeled using Oregon Green 488 (OG488). In detail, 5 mg TCL was mixed with 1 mg OG488 carboxylic acid, succinimidyl ester, 5-isomer (Thermo Fisher Scientific, O6147), and 1 mg OG488 maleimide (Thermo Fisher Scientific, O6034) in a total of 5 mL 50 mM HEPES (pH 7.4) and incubated overnight at 4 °C. To purify OG488-labeled TCL (OG488-TCL), the free OG488 carboxylic acid, succinimidyl ester, 5-isomer, and OG488 maleimide were removed using the PD-10 desalting columns (Cytiva, 17085101); 50 mM HEPES (pH 7.4) was used as the elution buffer.

B16F10-OVA melanoma cell line was engineered to express ovalbumin based on the B16F10 cell line (ATCC, CRL-6475)[70]. No authentication of this cell line was performed in this current study. The culture media for B16F10-OVA cells were prepared using Dulbecco's modified Eagle's medium (DMEM; Corning, 10-013-CV) supplemented with 10% (v/v) FBS, and 1% (v/v) Pen/Strep. Similar to ID8-Defb29/Vegf-a-Luc, B16F10-OVA cells were culture at 37 °C with 5% CO₂. For tumor inoculation, cells were first harvested using trypsin-EDTA, washed 3 times, and then resuspended in PBS at the required concentrations. For the ELISpot experiment, B16F10-OVA cells were harvested using the Cell Dissociation Buffer (Thermo Fisher Scientific, 13151014), washed 3 times using PBS, and then resuspended in the CTL media prior to experiments.

**Production of TCL-Lip and OVA-Lip.** TCL-Lip and ovalbumin (OVA)-Lip were produced using the thin-film rehydration method[71]. First, 410 μL 25 mg/mL DOPC (Avanti Polar Lipids, 850375C) and 200 μL 20 mg/mL DPPE-PEG2K-azide (Avanti Polar Lipids, 880231P) lipids in chloroform were mixed in a 10 mL round bottom flask to give a 10% molar ratio of azide groups. After removing the chloroform using a rotary evaporator at room temperature for 30 min, 5.13 mL 0.5 mg/mL TCL or OVA (Sigma-Aldrich, A2512) in 50 mM HEPES (pH 7.4) was added to rehydrate the lipid layer using the rotary evaporator at room temperature for 30 min, giving 1 mg TCL or OVA per 5 mg lipids. After rehydration and five freeze–thaw cycles as above, TCL-Lip and OVA-Lip were extruded 10 times using a GJE-10-mL jacketed liposome extruder (Genizer) and Whatman Nuclepore track-etched membranes with pore sizes of 200 nm (Cytiva, 10417006) and 100 nm (Cytiva110605). After the final extrusion, unpackaged TCL and OVA were removed by TFF using a MicroKros hollow-fiber filter (Repligen, C02-E500-10-N) with a 500-kDa cutoff and five washes in 50 mM HEPES (pH 7.4) until no protein was detected in the FT fraction using a BCA assay and UV–Vis. After the final TFF cycle, the volume of TCL-Lip and OVA-Lip was reduced to ~1 mL. The lipid concentration was determined using a phosphatidylcholine assay kit (Sigma-Aldrich, MAK049), and the concentration of packaged TCL and OVA was determined using a BCA assay (after mixing the liposomes with 1% (w/v) SDS (Thermo Fisher Scientific, 15553027) to break up the liposomes). OG488-TCL-Lip was prepared using the same method.

**Conjugation of CPMV to TCL-Lip and OVA-Lip.** We mixed 1 mg/mL CPMV-DBCO and 0.3 mg/mL TCL in TCL-Lip in a total of 1 mL for 8 h at room temperature to form the conjugated TCL-Lip–CPMV complex. The unconjugated mixture (TCL-Lip + CPMV) was prepared by mixing 1 mg/mL CPMV with 0.3 mg/mL TCL in TCL-Lip. OVA-Lip–CPMV and OVA-Lip + CPMV were prepared in the same manner. OG488-TCL-Lip-

CPMV-Cy5 was prepared by mixing 1 mg/mL CPMV-DBCO-Cy5 and 0.3 mg/mL OG488-TCL in liposomes. The OG488-TCL-Lip + CPMV-Cy5 mixture was prepared by mixing 1 mg/mL CPMV-Cy5 and 0.3 mg/mL OG488-TCL-Lip in liposomes.

## Characterization of CPMV, TCL, and liposomes as well as their conjugates

**NuPAGE**. To prepare loading samples for NuPAGE, all samples were first mixed with the NuPAGE LSD Sample Buffer (4X) (Thermo Fisher Scientific, NP0007) and NuPAGE Sample Reducing Agent (10X) (Invitrogen, NP0009) and then heated at 95 °C for 8 min. 20 μL of the prepared samples was then loaded into each lane of a 4–12% NuPAGE Bis-Tris Mini Protein Gel (Thermo Fisher Scientific, NP0321BOX). All gels were run at a constant voltage of 200 V and 25 W for 35 min using the MOPS SDS Running Buffer (Thermo Fisher Scientific, NP0001). Gels were then imaged using a ProteinSimple FluorChem R imager. For all samples that require fluorescent imaging, gels were first imaged using the MultiColor green filter to detect OG488 signals and the MultiColor red filter to detect Cy5 signals. Gels were then stained by Coomassie Brilliant Blue to image protein bands.

**UV–Vis spectrophotometry**. The concentrations of CPMV samples were determined by measuring the absorbance at 260 nm using a NanoDrop 2000 spectrophotometer (Thermo Fisher Scientific). After recording the absorbance of all samples, the concentrations were calculated using the CPMV extinction coefficient ($\varepsilon$) at 260 nm: 8.1 mL/(mg × cm). To determine the number of conjugated Cy5 per CPMV for CPMV-Cy5 or CPMV-DBCO-Cy5, the absorbance at 647 nm for Cy5 was recorded and the concentration of the conjugated Cy5 molecules was determined using the Cy5 molar extinction coefficient ($\varepsilon$): 271,000 L/(mol × cm). The flow-through (FT) fractions produced by TFF were also examined by spectrophotometry to ensure the complete removal of unpackaged TCL from TCL-Lip.

**Cy5 fluorescence intensity measurement**. We compared the intensity of CPMV-Cy5 and CPMV-DBCO-Cy5 fluorescence by measuring the emission of 1 mg/mL CPMV-Cy5 and CPMV-DBCO-Cy5 on a Tecan plate reader (excitation wavelength 633 nm, emission wavelength of 665 nm).

**Agarose gel electrophoresis**. All experiments were carried out using 1.2% (w/v) agarose gels. All gels were prepared using 1x Tris–Acetate EDTA (TAE) buffer (Thermo Fisher Scientific, B49) with 1x GelRed Nucleic Acid Gel Stain (Gold Biotechnologies, G-720-500). All loaded samples were prepared by mixing 10 μg CPMV and/or 30 μg TCL in liposomes with 1x Gel Loading Dye, Purple (New England Biolabs, B7024S), then loaded into each lane of the gel. All gels were run with a constant voltage of 100 V and a fluctuating current with 400 mA as the limit for 40 min in 1x TAE buffer. Gels were first imaged by UV to record the RNA; OG488, Cy5 signals, and the protein bands were then imaged similarly as the NuPAGE above.

**Dynamic light scattering (DLS)**. To measure the sizes of CPMV, liposomes, and their conjugates, we performed DLS measurements using a Zetasizer Nano ZSP/Zen5600 instrument (Malvern Panalytical). For each measurement, we loaded 100 μL of each sample containing 1 mg/mL of CPMV or liposomes containing 0.3 mg/mL of TCL or OVA into a disposable cuvette and measured each sample 3 times at room temperature.

**Transmission electron microscopy (TEM)**. All CPMV samples were first diluted to 0.5 mg/mL using 50 mM HEPES (pH 7.4). To prepare grids for TEM, Carbon Film 300 Mesh, Cu grids (Electron Microscopy Sciences, CF300-Cu-50) were first glow-discharged. After, 5 μL of each sample was applied on each grid, followed by a washing step using

Milli-Q water. To stain the protein samples, 5 μL uranyl acetate (1% (w/v), Electron Microscopy Sciences, 22400) was applied on each grid for 30 s. Uranyl acetate was lastly removed by filter paper and grids were air-dried for imaging. All grids were imaged using Talos transmission electron microscope (Thermo Fisher Scientific) and images were recorded at a nominal magnification of ×120,000.

**Cryogenic electron microscopy (Cryo-EM)**. Before loading the samples, 200-mesh copper TEM grids with lacey carbon film (Electron Microscopy Sciences, LC200-Cu) were glow-discharged using a Quorum Emiteck K100x device. We then applied 4 μL of each sample onto the grids using an FEI Vitrobot Mark III (Thermo Fisher Scientific). The samples were visualized using a Talos Arctica microscope (Thermo Fisher Scientific) and images were collected at a nominal magnification of ×92,000.

**Size exclusion chromatography (SEC)**. To perform SEC experiments, first, we diluted all samples to 1 mg/mL CPMV and/or liposomes with 0.3 mg/mL TCL. 100 μL of each sample was then analyzed using an AKTA purifier system (GE Healthcare) coupled with a Superose 6 Increase 10/300 GL. All experiments were run at a flow rate of 0.5 mL/min using 50 mM HEPES (pH 7.4) as the elution buffer and the absorbance at 280 nm was recorded.

## Mouse models

**Ovarian cancer efficacy study**. Five treatment groups ($n = 8$) were established: HEPES, CPMV, TCL-Lip, TCL-Lip + CPMV, and TCL-Lip−CPMV. All mice received six i.p. doses at weekly intervals on days −3, 4, 11, 18, 25 and 32. Each dose comprised 100 μg CPMV and/or 30 μg TCL in liposome in 200 μL 50 mM HEPES (pH 7.4). On Day 0, we injected $2 \times 10^6$ ID8-Defb29/Vegf-a-Luc cells in 200 μL PBS i.p. into each mouse. Mice were monitored every 2 days, and the tumor burden was recorded as the increase in body weight and circumference. All mice that survived over 100 days without any signs of tumor burden were treated as survivors who successfully rejected tumor growth. Mice were euthanized when their body weight reached 35 g or their circumference reached 9 cm.

**Dermal melanoma prevention study**. Five vaccination groups ($n = 10$ per group) were established: HEPES, CPMV, OVA-Lip, OVA-Lip + CPMV, and OVA-Lip−CPMV. All mice received two s.c. doses at bi-weekly intervals on days −28 and −14. Each dose comprised 100 μg CPMV and/or 30 μg OVA in liposome in 100 μL 50 mM HEPES (pH 7.4). On Day 0, we injected $2 \times 10^5$ B16F10-OVA cells in 20 μL PBS intradermally (i.d.) into the right flank of each mouse to establish dermal melanomas. Mice were monitored every 2 days, and the tumor volume was recorded. Tumor volumes were calculated using the formula $v = l \times \frac{w^2}{2}$, where $l$ is the length of the tumor and $w$ is the width. All mice free of tumors 50 days post-tumor inoculation were treated as survivors. All survivors were rechallenged by i.d. injecting $2 \times 10^5$ B16F10-OVA cells in 20 μL PBS into the left flank and age-matched mice ($n = 6$) were used as control. Mice were euthanized when their tumor volumes reached 1000 mm³. At the end of the study, all spleens were harvested for ELISpot and flow cytometry studies.

**Dermal melanoma surgery model and lung metastasis study**. Only the OVA-Lip−CPMV group ($n = 9$) and HEPES control group ($n = 10$) were studied. On day 0, $2 \times 10^5$ B16F10-OVA cells in 20 μL PBS were i.d. injected into the right flank of each mouse to establish dermal melanomas and on day 7, all tumors were surgically removed. Three doses of OVA-Lip−CPMV containing 100 μg CPMV and 30 μg OVA in 100 μL 50 mM HEPES (pH 7.4) were injected on days 7, 14, and 21; HEPES served as controls. On day 28, all mice were challenged with tail vein intravenously (i.v.) injected $1 \times 10^5$ B16F10-OVA in 100 μL PBS to establish lung metastasis. Mice were sacrificed on day 63 to collect

lungs. Harvested lungs were stored in 10% (v/v) neutral-buffered formalin (Millipore Sigma, HT501128-4L) for 24 h followed by 70% (v/v) ethanol to check lung metastasis. Processed lungs were then sent to UCSD Tissue Technology Shared Resources-Histology Core for H&E toxicology analysis.

**Analysis of bone marrow-derived dendritic cells (BMDCs).** BMDCs were prepared from a single-cell suspension of whole bone marrow cells isolated from the femurs and tibias of five female C57BL/6J mice. The cells were washed once with PBS, and the red blood cells were lysed for 5 min at room temperature using RBC lysis buffer (Invitrogen, 00-4300-54). The cells were washed twice with PBS, then resuspended in T-cell medium (RPMI 1640 medium + L-glutamine supplemented with 10% (v/v) FBS, 1% (v/v) Pen/Strep, 1 mM sodium pyruvate and 0.05 mM β-mercaptoethanol). To differentiate BMDCs, $3 \times 10^6$ cells per well were seeded into Costar TC-treated 6-well plates (Corning, 3516). The medium was supplemented with 10 ng/mL mouse IL-4 (Biolegend, 574304) and 15 ng/mL mouse GM-CSF (Biolegend, 576304) and cultured for 6 days at 37 °C in a 5% $CO_2$ atmosphere. On day 4, fresh medium was added containing 10 ng/mL mouse IL-4 and 15 ng/mL mouse GM-CSF. After 6 days, BMDCs were harvested and adjusted to $1 \times 10^6$ cells/mL in fresh medium and seeded into 12-well plates (Corning, 3513) with $2 \times 10^6$ cells/well. Treatments containing 1 mg/mL CPMV-Cy5 and/or 0.3 mg/mL OG488-TCL and 50 mM HEPES (pH 7.4) control were added to the BMDCs and incubated as above for 1 or 24 h. The cells were then harvested for flow cytometry and confocal microscopy. For flow cytometry, BMDCs were blocked using anti-CD16/32 Fc block (1:500, Biolegend, 101302) on ice for 30 min, then stained using a PE-conjugated anti-CD11c antibody (1:100, Biolegend, 117308) on ice for 1 h and fixed using the Stabilizing Fixative 3x Concentrate (BD Biosciences, 338036). After washing, BMDCs were resuspended in BD Pharmingen Stain Buffer (BD Biosciences, 554657) and analyzed using a BD Accuri C6 Plus flow cytometer (BD Biosciences). Data were analyzed using Flowjo_v10.7. For confocal microscopy, leftover BMDCs from flow cytometry experiments were centrifuged onto Superfrost Plus Microscope Slides (Fisher Scientific, 22-037-246) using a Statspin Cytofuge Cytocentrifuge (Beckman Coulter). All slides were then stained and mounted using Fluoroshield with DAPI (Millipore Sigma, F6057). Fluorescence images were obtained using a Nikon A1R confocal microscope with an Apo TIRF 100×/1.49 oil objective (Nikon). Collected images were analyzed using NIS-Elements AR Analysis v5.30 (Nikon).

**Lymphocytes uptake experiments.** To study the co-delivery of TCL and CPMV into draining lymph nodes, treatments containing 100 μg CPMV-Cy5 and/or 30 μg OG488-TCL in 50 μL 50 mM HEPES (pH 7.4) and 50 mM HEPES (pH 7.4) control were injected s.c. into the footpad of female C57BL/6J mice, $n = 5$ per group. After 4, 8, 24, 72 and 168 h, one mouse from each group was sacrificed to collect the two PDLNs for fluorescence imaging using a Xenogen IVIS 200 imaging system to quantify OG488 and Cy5 fluorescence. The collected PDLNs were then processed to prepare slides for immunofluorescence imaging. In detail, PDLNs were flash frozen in Tissue-Tek OCT medium (Sakura, 4583) under liquid nitrogen, then 10-μm sections were prepared using a Leica CM1860 Cryostat. Sections were fixed in 4% (v/v) paraformaldehyde (Electron Microscopy Sciences, 15700) at room temperature for 10 min, then washed with PBS and blocked with 1% (w/v) bovine serum albumin (BSA) (Fisher Scientific, BP671-1) for 1 h at room temperature. DCs, macrophages, B cells, and T cells were stained with the following primary antibodies in PBS containing 1% (w/v) BSA at 4 °C overnight: Armenian hamster anti-mouse CD11c monoclonal antibody (1:100, Biolegend, 117302), rat anti-mouse F4/80 monoclonal antibody (1:100, Biolegend, 123110), rat anti-mouse B220 monoclonal antibody (1:100, Biolegend, 103202), and rat anti-mouse CD3 monoclonal antibody (1:100, Biolegend, 100202).

The next day, DCs were stained using a goat anti-Armenian hamster TRITC polyclonal antibody (1:500, Abcam, ab5741), and the other cells were stained using a goat anti-rat Alexa Fluor 555 polyclonal antibody (1:500, Invitrogen, A-21434) at room temperature for 1 h. After washing with PBS and drying, the sections were mounted with Fluoroshield containing DAPI and imaged using a Nikon A1R confocal microscope with a filter set (DAPI, TRITC, FITC and APC) and a ×20 objective lens. All images were then processed using NIS-Elements AR Analysis v5.30. Single lymphocytes were prepared from the harvested PDLNs using the Spleen Dissociation Kit (Miltenyi Biotec, 130-095-926) and a gentleMACS Octo Dissociator with heaters (Miltenyi Biotec). They were blocked, stained and fixed in preparation for flow cytometry as described above.

**Lymphocytes activation and presentation experiments.** To study the activation of DCs in PDLNs, treatments containing 100 μg CPMV and/or 30 μg TCL and 50 mM HEPES (pH 7.4) control were injected in the left and right footpad of female C57BL/6J mice ($n = 3$ per group) and the PDLNs were processed as described above. Lymphocytes were then stained using LIVE/DEAD Fixable Aqua Dead Cell Stain Kit (Thermo Fisher Scientific, L34957) on ice for 30 min to identify live cells. After washing, lymphocytes were blocked as above then stained with Pacific Blue anti-CD45 antibody (1:100, Biolegend, 103126), Super Bright 780 anti-CD11c antibody (1:100, Thermo Fisher Scientific, 78-0114-82), APC MHC-II antibody (1:100, Thermo Fisher Scientific, 17-5320-82), FITC anti-CD40 antibody (1:100, Thermo Fisher Scientific, 11-0402-82), Brilliant Violet 605 anti-CD80 antibody (1:100, Biolegend, 104729), and PerCP-Cy5.5 anti-CD86 antibody (1:100, Biolegend, 105028) on ice for 1 h. UltraComp eBeads (Thermo Fisher Scientific, 01-2222-42) were used to prepare single-color staining for compensation. Lymphocytes were then fixed, washed, and resuspended for flow cytometry as described above. Flow cytometry was performed using a BD FACSCelesta™ Cell Analyzer (BD Biosciences).

To study antigen presentation on DCs in PDLNs, TCL was replaced with the model antigen OVA to formulate OVA-Lip–CPMV and other groups. 100 μg CPMV and/or 30 μg OVA and 50 mM HEPES (pH 7.4) control were injected into the left and right footpad of female C57BL/6J mice ($n = 3$ per group) and the PDLNs were collected at 48 h, then processed as described above. Lymphocytes were then stained using LIVE/DEAD Fixable Green Dead Cell Stain Kit (Thermo Fisher Scientific, L34970) on ice for 30 min to identify live cells. After washing, lymphocytes were blocked as above and then stained with APC anti-CD11c antibody (1:100, Thermo Fisher Scientific, 17-0114-82), PE anti-mouse H-2K$^b$ bound to SIINFEKL antibody (1:100, Biolegend, 141604) on ice for 1 h. Lymphocytes were then fixed, washed, and resuspended for flow cytometry using a BD FACSCelesta™ Cell Analyzer.

**Assessment of CD4$^+$ and CD8$^+$ T cells for TCL-Lip–CPMV vaccination.** To evaluate T-cell modulation within the i.p. space and spleen, treatments and control were administered as described above for the efficacy study ($n = 5$ per group). Seven days after the third injection, mice were sacrificed and i.p. washes and spleens were collected from five mice per group. Spleens were processed into single-cell splenocytes using the Spleen Dissociation Kit and a gentleMACS Octo Dissociator with heaters. The red blood cells in the samples were lysed using RBC lysis buffer and the remaining cells were prepared using the LIVE/DEAD Fixable Aqua Dead Cell Stain Kit as described above. After washing and blocking as described above, cells were stained with Pacific Blue anti-CD45 antibody (1:100, Biolegend, 103126), APC-Cy7 anti-CD3 antibody (1:100, Biolegend, 100222), FITC anti-CD4 antibody (1:100, Biolegend, 100406), Brilliant Violet 785 anti-CD8 antibody (1:100, Biolegend, 100750), Brilliant Violet 605 anti-CD44 antibody (1:100, Biolegend, 103047), APC anti-CD62L antibody (1:100, Biolegend, 104412), and PerCp-Cy5.5 anti-CD69 antibody (1:100, Biolegend, 104522). UltraComp eBeads were used to prepare single-color staining

for compensation. The cells were fixed, washed, and resuspended for flow cytometry as described above.

**Assessment of antigen-specific CD8+ T cells.** TCL was replaced with the model antigen OVA and the corresponding treatments containing 100 µg CPMV and/or 30 µg OVA were injected i.p. into C57BL/6 J mice (n = 5 per group) in a prime-boost regimen on days 0 and 7; 50 mM HEPES (pH 7.4) as control. On day 14, the mice were sacrificed and i.p. washes and spleens were harvested and processed as described above. The cells were further prepared according to the MHC-I Dextramer Staining Protocol. Briefly, the cells were stained with the PE-conjugated H-2 Kb SIINFEKL Dextramer (1:100, Immudex, JD02163) for 10 min on ice, then with Pacific Blue anti-CD45 antibody (Biolegend), APC-Cy7 anti-CD3 antibody (1:100, Biolegend, 100222), FITC anti-CD4 antibody (1:100, Biolegend, 100406) and Brilliant Violet 785 anti-CD8 antibody (1:100, Biolegend, 100750) for an additional 20 min. After five washes, the cells were resuspended for analysis by flow cytometry as described above.

**ELISpot assay of the B16F10-OVA dermal melanoma rechallenge study.** Spleens from the B16F10-OVA rechallenge study (Fig. 6f) (6 for HEPES control group, 6 for OVA-Lip–CPMV group, and 1 for OVA-Lip group) were harvested by the end of the study and processed into single-cell splenocytes using the same protocol above. All splenocytes were resuspended in CTL media at $5 \times 10^6$ cells/mL. Following the manufactural protocol, the ELISpot assay was carried out using a Mouse interferon (IFN)-γ single color ELISPOT kit with a 96-well plate (Cellular Technology Limited, mIFNgIL4-2M/2). Simply, 500,000 splenocytes were plated into each well of the 96-well plate and then stimulated with 20 µg/mL OVA 257-264 (SIINFEKL) peptide (InvivoGen, vac-sin) or 500,000 B16F10-OVA cells for 48 hours at 37 °C and 5% CO₂. Post the incubation, the plates were processed to develop IFN-γ spots. The colored spots were quantified using an Immunospot S6 Entry analyzer (Cellular Technology Limited).

**Flow cytometry of B16F10-OVA stimulated splenocytes.** Splenocytes (as prepared for the ELISpot assay) were analyzed by flow cytometry to examine whether antigen-specific CD8 T cells were produced. Splenocytes were exposed to B16F10-OVA cells and analyzed for production of IFN-γ, Granzyme B, and Perforin within CD8 T cells using a BD Cytofix/Cytoperm™ Plus Fixation/Permeabilization Solution Kit with BD GolgiStop™ (BD Biosciences, 554715). $1 \times 10^6$ splenocytes were mixed with $1 \times 10^6$ B16F10-OVA cells in a total of 0.5 mL RPMI 1640 medium + L-glutamine supplemented with 10% (v/v) FBS, 1% (v/v) Pen/Strep, plated in one well of a 48-well plate, and incubated for 48 hours at 37 °C and 5% CO₂. 3 hours prior to the end of the incubation, 0.5 µL BD GolgiPlug was added into each well to trap intracellular IFN-γ, Granzyme B, and Perforin. Post 48 h incubation, cells were harvested and stained for flow cytometry using the protocol described above. The cells were stained first by LIVE/DEAD Fixable Aqua Dead Cell Stain Kit, followed by anti-CD16/32 Fc block, Pacific Blue anti-CD45 antibody, APC-Cy7 anti-CD3 antibody, FITC anti-CD4 antibody, and Brilliant Violet 785 anti-CD8 antibody surface staining. Then cells were fixed and permeabilized using the BD Fixation/Permeabilization solution for 20 min on ice, washed twice using BD Perm/Wash™ buffer, then stained using PerCp-Cy5.5 anti-IFN-γ antibody (1:100, Invitrogen, 45-7311-82), APC anti-Granzyme B antibody (1:500, Invitrogen, 17-8898-82), and PE anti-Perforin antibody (1:100, Invitrogen, 12-9392-82) for 30 min on ice. Cells were then washed and resuspended for flow cytometry analysis using a BD FACSCelesta™ Cell Analyzer.

**Biodistribution and biosafety experiments.** To evaluate the biodistribution and clearance of injected vaccines, 1 dose of fluorescently labeled TCL-Lip–CPMV, TCL-Lip + CPMV, and 50 mM HEPES (pH 7.4) control were i.p. injected into mice (3 mice per group). 24 h and 7 days later, mice were sacrificed and i.p. washes, blood, and organs (brain, lung, heart, liver, kidney, and spleen) were collected. Blood and i.p. washes were centrifuged at 5283×g for 10 min at 4 °C to collect the plasma and i.p. wash supernatant. Organs were weighted and homogenized within 1 mL PBS using a Cole-Parmer LabGEN 125 homogenizer. Supernatant of all organs was collected by centrifuging at 8000 g for 5 min. OG488 (TCL) and Cy5 (CPMV) fluorescence for the i.p. washes, blood plasma, and organ supernatant were measured using a Tecan Infinite M200 plate reader (OG488: excitation 498 nm, emission 526 nm, gain 120; Cy5: excitation 646 nm, emission 662 nm, gain 55).

To evaluate the toxicity of TCL-Lip–CPMV and TCL-Lip + CPMV, both samples were i.p. injected into mice. 24 h and 7 days later, mice were sacrificed to collect organs (brain, lung, heart, liver, kidney, and spleen). All organs were sent to UCSD Tissue Technology Shared Resources-Histology Core for H&E toxicology analysis.

### Software for figure generation
CPMV structure is generated using UCSF Chimera 1.16 using PDB: 1NY7. Chemical structures are generated using ChemDraw 20.0.0. TCLs-Lip–CPMV and OVA-Lip–CPMV are generated using BioRender. Treatment and injection schedule figures are generated using BioRender and Adobe Illustrator 2023. Histology images are processed using Aperio ImageScope 12.4.6. Other software includes NIS-Elements AR Analysis v5.30 (Nikon), GraphPad Prism 10.3.1, and FlowJo 10.7.1.

### Reporting summary
Further information on research design is available in the Nature Portfolio Reporting Summary linked to this article.

## Data availability
All data generated in this study are available within the Article, Supplementary Information, and Source Data file. Raw data are available from the corresponding author upon request. Source data are provided with this paper.

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

## Acknowledgements

This work was supported in part by an NIH R01-CA253615 grant to N.F.S., the Shaughnessy Family Fund for Nano-ImmunoEngineering (nanoIE) at UCSD to N.F.S., an NCI K99-CA283299 to Z.Z., and the Kirschstein National Research Service Award—Cancer Therapeutics Training (CT2) program at the UCSD Moores Cancer Center to D.L. We thank the Nano3 core facility at UCSD for TEM imaging. Nano3 is the San Diego Nanotechnology Infrastructure (SDNI) of the UCSD, a member of the National Nanotechnology Coordinated Infrastructure (NNCI), which is supported by the National Science Foundation (Grant ECCS-1542148). We thank the Cryo-EM core facility at UCSD for cryo-EM imaging, the UCSD Cancer Center Microscopy Shared Facility (Specialized Support Grant P30 CA23100-28), and UCSD Cancer Center Support Grant 2P30CA023100 by NCI for IVIS imaging and confocal imaging. We thank the UCSD Tissue Technology Shared Resource for histology studies; the core is supported by a National Cancer Institute Cancer Center Support Grant (CCSG Grant P30CA23100). We thank the La Jolla Institute Flow Cytometry Core for flow cytometry experiments. We thank Professor Liang-fang Zhang at UCSD for sharing the B16F10-OVA cell line.

## Author contributions

Z.Z. and N.F.S. designed the experiments. Z.Z., D.L., J.F.A.O., and A.O.O. performed the experiments. Z.Z., D.L., J.F.A.O., A.O.O., and N.F.S. analyzed the data, and Z.Z. and N.F.S. wrote the manuscript.

## Competing interests

N.F.S. is a co-founder and CEO of and has equity in PlantiosX Inc. She is also a co-founder of and has equity in Mosaic ImmunoEngineering Inc. and co-founder and manager of Pokometz Scientific LLC, under which she is a paid consultant to Flagship Labs 95 Inc. The other authors declare no competing interests.
