## [Transparent Peer Review file · Nature Communications]

A cowpea mosaic virus adjuvant conjugated to liposomes loaded with tumor cell lysates as an ovarian cancer vaccine

Corresponding Author: Professor Nicole Steinmetz

Version 0:

Reviewer comments:

Reviewer #1

(Remarks to the Author)

In this manuscript Zhao et al., present a study of the vaccine adjuvant cowpea mosaic virus (CPMV) co-delivered with tumour cell lysate as a source of antigen. They demonstrate anti-tumour effect in a murine model of ovarian cancer. They go on to examine uptake of nanoparticles and changes in immune cells post treatment, particularly DCs and T cells. They use OVA as a model antigen to assess presentation by APCs and the generation of specific T cell responses. The nanoparticles are characterised and potential toxicity of the therapy is assessed.

Fig 1. The characterisation of the nanoparticle seems appropriate

Fig 2. The labelling system used to track components and uptake (in vitro) is well thought out.

Fig 3. The confocal images (and in supps) are somewhat hard to interpret due to clarity of cellular staining (eg. CD11c), but the data is backed up by quantification using flow cytometry and presentation of a model antigen is assessed demonstrating functional presentation of an antigen in the vaccine construct.

Figure 4. The authors examine activation (CD69) and memory in T cells (CD44/CD62L). While there is some improvement in T cell activation in the combination, most of the effect appears to come from the CPMV. Memory populations are expanded non-specifically by the therapy as there is no further increase in combination group over CPMV alone, which is surprising. Antigen/vaccine-specific T cells are not identified or characterised.

Figure 5. Here antigen specific T cell expansion is measured after vaccination using the OVA model antigen, proving good expansion of antigen specific CD8 T cell vaccination. The gating in 5c can probably be adjusted to more accurately capture the CD8 population, which may even decrease the background further (as is done in Supp Figure 22). This analysis should be consistent.

Figure 6. Shows the effect on tumour growth in a prophylactic setting, rather than a model of proper relapsed or metastatic disease as is mentioned in the abstract. There are clearly some responders to the TCL vaccine, which is improved with the TCL-Lip CMV preparation. However, CPMV on its own has a strong and non-specific (??) anti-tumour effect, so it is hard to disentangle the improvement to and mechanism of the antigen/vaccine-specific response

Figure 7. Examines biodistribution of vaccine components suggesting spread to several organs after i.p. injection, but what about intra muscular or intra-tumoral administration? H&E analysis shows no sign of toxicity at the given dose.

Overall the authors have done a considerable amount of work, creating a combination nanoparticle delivery system for tumour antigens and adjuvant. They have analysed whether addition of vaccine adjuvant CPMV to the TCL vaccine improves presentation of antigen and downstream T cell phenotype. They demonstrate potential for improved anti-tumour response but only in a single prophylactic model.

Some points that could be addressed to improve the impact of the manuscript are:

Use of a second tumour model, potentially one which uses ex vivo tumour samples to create lysate or better models the setting of relapsed or metastatic disease e.g. removal or debulk of primary tumour as a source of TCL and treatment of residual disease or metastatic spread.

I am unsure if antigen-specific responses to the tumour cell line used can be identified. Otherwise, using a tumour cell line modified to express a model antigen (such as OVA) may be useful. This would also be a better source of antigen than simply loading liposomes with OVA as was done for Figure 5, although this approach does yield clear positive results.

The use of the OVA system opens up the possibility to look at the phenotype and particularly the effector function of vaccine-induced T cells in more depth. This was somewhat lacking in the flow cytometry analysis, which only assessed the whole T cell compartment (CD4/CD8). This hampers a deeper interpretation of the antigen-specific vs. non-specific effects of the vaccine construct as a whole.

Some minor points:

Supp Figure 20 - Gating should be adjusted for clarity.

Figure 3 - Sub-figure (h) is missing a description in the figure legend. Y-axis lists CD11b+ DC but the text reads CD11c+.

Reviewer #2

(Remarks to the Author)

The revised manuscript by Zhao et al. has addressed some of the concerns raised in initial review. However, major concerns remain relating to the relevance of the model system used to evaluate ovarian cancer anti-tumour immunity induced by the TCL-Lip-CPMV system. The model used does not evaluate autologous tumour cell lysates from established tumours, the ID8 cell line itself is lacking the Trp53 mutation which is almost universally present in high-grade serous ovarian cancer and homologous recombination in ID8 is intact. No mutations typical of other types of ovarian cancer (eg clear cell, low-grade serous) have been found either. Processing and presentation has not been assessed for tumour antigens and the complexity of immunity to TCLs derived from established tumours cannot be evaluated. The ovalbumin system illustrates that the nanoparticles work for immunity against ovalbumin but this is only preliminary data for the evaluation of antigen-specific anti-tumour immune response, similar in-depth data for which is not provided. In the clinical setting women with ovarian cancer are diagnosed with advanced stage disease, with cancer already disseminated at the time of diagnosis. Administration of the viral constructs three days prior to malignant cell injection, does not replicate the setting in human disease, a more relevant design is needed to assess the efficiency of the construct as prophylactic for recurrent disease. Although the nanoparticle system is comprehensive, the study in its current form does not provide sufficiently strong evaluation of this technology in anti-tumour immunity in ovarian cancer.

Reviewer #3

(Remarks to the Author)

The manuscript entitled "A cowpea mosaic virus adjuvant conjugated to liposomes loaded with tumor cell lysates as an ovarian cancer vaccine" described a cancer nanovaccine composed of TCL, liposomes, and CPMV.

First of all, the novelty of this kind nanovaccine is limited as pointed by reviewer 1.

Second, the challenged tumor models, such as established subcutaneous tumor models or relapse tumor models post surgery, could be used to evaluate the efficacy potency of the nanovaccine. BTW, in the current treatment model, the mice were administrated by 6 doses of TCL-Lip-CPMV, which were rarely used in the study of cancer vaccines (normally, 2~3 doses).

At last, the immunology mechanism of CPMV as adjuvant and safety profile study could be strengthened. For example, adding another adjuvant (e.g. CPG or Pam 3) as control, detection of tumor specific CD8+ T cell in peripheral blood or LNs, detection of AST and ALT.

Reviewer #4

(Remarks to the Author)

The author has already satisfactorily addressed my question and it can be published in its current form.

Version 1:

Reviewer comments:

Reviewer #1

(Remarks to the Author)

The authors have completed additional experiments using the B16.OVA tumour model to look at antigen-specific tumour responses. This strengthens the manuscript and brings it in line with other similar scientific papers reporting on cancer/vaccines.

While looking at a model tumour antigen allows for clear examination of antigen-specific responses and prophylactic treatment experiments. Perhaps a better way to complete these studies would have to use tumour lysate from the B16 cell line and assay e.g. with ELISPOT, several of the endogenous tumour antigens which have been reported in the literature for the B16 cells. As these often fall somewhat short of the results obtainable using OVA and this method would be more in line with the authors previous strategy of using TCL.

The use of orthotopic (i.d.) melanoma engraftment has been reported to lead to metastasis to local lymph nodes and other organs once tumours become large. The surgical model employed in the revisions is somewhat unusual as it involves surgical removal of a small primary (d7), most likely prior to any met formation, followed by vaccination and then artificial seeding of lung mets using the i.v. inoculation route (A common prophylactic approach). In this scenario, I wonder why the earlier surgery needs to be performed and what effect the authors think it might be having on the vaccination. Presumably there is some endogenous response to the first tumour that the vaccination can boost? Where they setting the effect of prior surgery on the vaccination? However, this is not really representative of surgical resection with residual primary or metastatic disease, developing alongside the primary tumour.

Reviewer #2

(Remarks to the Author)

The authors have conducted additional work that has improved several parts of the manuscript and demonstrated vaccine efficacy in a surgical model of B16F10-ovalbumin expressing melanoma tumours. However, my concern remains on the data supporting the efficacy of TCL_Lip_CPMV complexes in ovarian cancer as the manuscript still relies solely on the ID8-Defb29/Vegf-a-Luc model. This model has several critical limitations as an ovarian cancer model, including the expression of the weakly immunogenic firefly luciferase which, in the context of CPMV, could be driving the observed outcomes.

Reviewer #3

(Remarks to the Author)

I continue to harbor reservations regarding the efficacy of the nanovaccine. Specifically, the experimental design appears unconventional, as it involves administering a single injection prior to tumor inoculation. In clinical practice, it is impractical to develop personalized cancer vaccines before tumors are detected. Despite this unusual setup, the statistical analysis revealed no significant differences between the TCL-Lip + CPMV group and the TCL-Lip-CPMV or CPMV-only groups ($p > 0.05$), as illustrated in Figures 6a and 6b. Similarly, the new B16F10-OVA model failed to provide novel insights into the vaccine's efficacy, particularly in the context of ovarian cancer. Finally, while the treatment model depicted in Figures 6k-n did demonstrate some potential efficacy compared to the control, the appropriate control group should have been TCL-Lip-CPMV or CPMV-only, rather than HEPES.

In summary, the authors have not provided sufficient evidence to substantiate the efficacy of TCL-Lip-CPMV as an ovarian cancer vaccine.

Version 2:

Reviewer comments:

Reviewer #1

(Remarks to the Author)

Regarding the authors response to my comments (R1):

I appreciate the updated rationale for the proposed nanovaccine as this helps to clarify the intended translational use case without downplaying the limitations of the study and models used throughout.

Considering the ELISPOT experiment, it would have been interesting to examine the response against both the B16.OVA-expressing cell line and parental B16 cell line in co-culture experiments to determine the extent (if any) of epitope spreading that is induced by the nanovaccine. I appreciate that these experiments used vaccinated mice as the source of splenocytes, making additional experiments difficult without using more animals. This data would be of interest in the context of cancer vaccination in patients, where dominant antigens are often the target of downregulation by tumours under immune pressure and critical e.g. in relapsed/metastasised cases.

Additionally, for Supp. Figure 26, why is there no OVA-Lip group in (c/d) and no OVA-lip-CPMV group in (e-g) for comparison. I apologise if I am missing something here.

Considering the surgical model of relapse/remission; in the clinical setting, can the authors confirm whether recurrence is predominantly local or via cryptic metastatic disease. The later case is modelled in their experimental design but not the setting of local relapse, which while difficult to test in the ovarian setting, could certainly have been modelled in the B16 flank

model or common breast cancer models.

In either scenario human patients will not be truly disease free during remission, as the mice are in this case, and this is likely to be important for vaccination i.e. via ongoing cancer-induced immunomodulation and as a source of additional antigen.

Reviewer #3

(Remarks to the Author)

While the authors have acknowledged certain limitations in the data quality regarding efficacy of the nanovaccine and offered explanations that may appear somewhat strained, they have not taken concrete steps to address these scientific concerns. On this basis, I remain unconvinced that sufficient evidence has been provided to substantiate the efficacy of TCL-Lip-CPMV as a viable ovarian cancer vaccine.

Dear Reviewers:

We appreciate your comments and questions. We have carefully reviewed all your points, and to address their suggestions thoroughly, we have carried out additional experiments. A point-by-point response is shown below. We have revised our manuscript, and the revisions are highlighted in yellow.

We thank you for time and hope you will find it acceptable for publication in Nature Communications.

Best wishes,

Nicole F. Steinmetz PhD
Professor, UC San Diego

REVIEWER COMMENTS

Reviewer #1 (Remarks to the Author):

In this manuscript Zhao et al., present a study of the vaccine adjuvant cowpea mosaic virus (CPMV) co-delivered with tumour cell lysate as a source of antigen. They demonstrate anti-tumour effect in a murine model of ovarian cancer. They go on to examine uptake of nanoparticles and changes in immune cells post treatment, particularly DCs and T cells. They use OVA as a model antigen to assess presentation by APCs and the generation of specific T cell responses. The nanoparticles are characterised and potential toxicity of the therapy is assessed.

Fig 1. The characterisation of the nanoparticle seems appropriate

Fig 2. The labelling system used to track components and uptake (in vitro) is well thought out.

Fig 3. The confocal images (and in supps) are somewhat hard to interpret due to clarity of cellular staining (eg. CD11c), but the data is backed up by quantification using flow cytometry and presentation of a model antigen is assessed demonstrating functional presentation of an antigen in the vaccine construct.

We thank the reviewer for the positive comments.

Figure 4. The authors examine activation (CD69) and memory in T cells (CD44/CD62L).

While there is some improvement in T cell activation in the combination, most of the effect appears to come from the CPMV. Memory populations are expanded non-specifically by the therapy as there is no further increase in combination group over CPMV alone, which is surprising. Antigen/vaccine-specific T cells are not identified or characterised.

Indeed, most of the effects are coming from CPMV in terms of T cell activation (and innate immune cell activation). Our previous work [1] and the present study consistently demonstrates that CPMV is a strong adjuvant and immunostimulator to modulate the tumor microenvironment resulting in T cell mediated anti-tumor immunity.

The TCL packaged in liposomes (without the CPMV adjuvant) did not activate DCs and T cells (Fig. 3e-g and Fig. 4), highlighting the need of an adjuvant. CPMV activates immune cells and primes T cells – when the TCL is co-delivered the pool of overall T cells is expanded (Fig. 4). Due to the complexity of TCL, it's impossible to characterize antigen-specific T cells for the TCL vaccines. Therefore, we have carried out experiments using ovalbumin to replace TCL. Our data (Fig. 3h) demonstrated that only CPMV conjugated with ovalbumin loaded liposomes yield more SIINFEKL peptides presented on DCs (Fig. 3h) and more antigen-specific T cells (SIINFEKL positive CD8 T cells) (Fig. 5) compared to all other groups, showing the importance of delivering both antigen and adjuvant to boost antigen presentation and to achieve antigen-specific T cells expansion.

[1] Wang, Chao, Steven N. Fiering, and Nicole F. Steinmetz. "Cowpea mosaic virus promotes anti-tumor activity and immune memory in a mouse ovarian tumor model." *Advanced therapeutics* 2.5 (2019): 1900003.

Figure 5. Here antigen specific T cell expansion is measured after vaccination using the OVA model antigen, proving good expansion of antigen specific CD8 T cell vaccination. The gating in 5c can probably be adjusted to more accurately capture the CD8 population, which may even decrease the background further (as is done in Supp Figure 22). This analysis should be consistent.

We thank the reviewer for the suggestion. We have updated both Fig. 5c and Supplementary Fig. 22b by plotting them consistently.

Figure 6. Shows the effect on tumour growth in a prophylactic setting, rather than a model of proper relapsed or metastatic disease as is mentioned in the abstract. There are clearly some responders to the TCL vaccine, which is improved with the TCL-Lip CMV preparation. However, CPMV on it's own has a strong and non-specific (??) anti-tumour effect, so it is hard to disentangle the improvement to and mechanism of the antigen/vaccine-specific response

To address the reviewer's concerns, we have completed a set of new experiments using the B16F10-OVA melanoma model as shown in the updated Fig. 6, Supplementary Fig. 25, and Supplementary Fig. 26– here we specifically address the

vaccine-specific response. In our new studies, we observed that conjugating CPMV to ovalbumin loaded liposomes significantly enhanced overall survival against the B16F10-OVA challenge (Fig. 6e). This treatment could also lead to effective immune protection for mice against the B16F10-OVA melanoma challenge (Fig. 6f-j). To investigate this vaccine's potency in a treatment setting, vaccination was administered after tumor removal surgery (Fig. 6k) and could prevent lung metastasis (Fig 6l-n).

Figure 7. Examines biodistribution of vaccine components suggesting spread to several organs after i.p. injection, but what about intra muscular or intra-tumoral administration? H&E analysis shows no sign of toxicity at the given dose.

We appreciate the review's suggestion for testing other routes of administration. As for ovarian cancer, we chose i.p. injections to achieve "intra-tumoral" injections. As for the newly added B16F10-OVA melanoma model, we chose s.c. injections to test out our vaccine hypothesis. As for our future research work, we will set to explore the intra-muscular administration of our vaccines.

Overall the authors have done a considerable amount of work, creating a combination nanoparticle delivery system for tumour antigens and adjuvant. They have analysed whether addition of vaccine adjuvant CPMV to the TCL vaccine improves presentation of antigen and downstream T cell phenotype. They demonstrate potential for improved anti-tumour response but only in a single prophylactic model.

We thank the reviewer for bringing up this concern and we have addressed this with a new set of studies using B16F10-OVA dermal melanoma and B16F10-OVA lung metastasis model post-surgery—the results are shown in the updated Fig. 6d-n, Supplementary Fig. 25, and Supplementary Fig. 26.

Some points that could be addressed to improve the impact of the manuscript are:

Use of a second tumour model, potentially one which uses ex vivo tumour samples to create lysate or better models the setting of relapsed or metastatic disease e.g. removal or debulk of primary tumour as a source of TCL and treatment of residual disease or metastatic spread.

We appreciate the reviewer's suggestion. We have carried out new experiments using the B16F10-OVA melanoma model. Specifically, we used ovalbumin as the tumor specific antigens to replace TCL and demonstrated exceptional efficacy in the prophylactic setup and the surgical setup as shown in the updated Fig. 6d-n, Supplementary Fig. 25, and Supplementary Fig. 26.

I am unsure if antigen-specific responses to the tumour cell line used can be identified. Otherwise, using a tumour cell line modified to express a model antigen (such as OVA) may be useful. This would also be a better source of antigen than simply loading liposomes with OVA as was done for Figure 5, although this approach does yield clear positive results.

We have completed these experiments as suggested using OVA as a model antigen; the data is shown in the updated Fig. 6d-n, Supplementary Fig. 25, and Supplementary Fig. 26.

The use of the OVA system opens up the possibility to look at the phenotype and particularly the effector function of vaccine-induced T cells in more depth. This was somewhat lacking in the flow cytometry analysis, which only assessed the whole T cell compartment (CD4/CD8). This hampers a deeper interpretation of the antigen-specific vs. non-specific effects of the vaccine construct as a whole.

We thank the reviewer's comment and have completed the studies as suggested using the B16F10-OVA tumor model. By combining with the flow cytometry study for the SIINFEKL specific T cells (Fig. 5), we could draw the conclusion that OVA-Lip-CPMV lead to enhanced SIINFEKL presentation on DCs (Fig. 3h), which ultimately resulted in increased number of SIINFEKL specific CD8 T cells (Fig. 5). The enhanced tumor antigen-specific CD8 T cell response correlates with enhanced efficacy and better protection against the antigen expressing tumors (B16F10-OVA) (see updated Fig. 6d-n, Supplementary Fig. 25, and Supplementary Fig. 26.).

Some minor points:

Supp Figure 20 - Gating should be adjusted for clarity.

We have adjusted the gating now as shown in the updated Supplementary Fig. 20.

Figure 3 - Sub-figure (h) is missing a description in the figure legend. Y-axis lists CD11b+ DC but the text reads CD11c+.

We have added the figure legend for Fig. 3h and corrected the label for Y-axis.

Reviewer #2 (Remarks to the Author):

The revised manuscript by Zhao et al. has addressed some of the concerns raised in initial review. However, major concerns remain relating to the relevance of the model system used to evaluate ovarian cancer anti-tumour immunity induced by the TCL-Lip-CPMV system. The model used does not evaluate autologous tumour cell lysates from established tumours, the ID8 cell line itself is lacking the Trp53 mutation which is almost universally present in high-grade serous ovarian cancer and homologous recombination in ID8 is intact. No mutations typical of other types of ovarian cancer (eg clear cell, low-grade serous) have been found either. Processing and presentation has not been assessed for tumour antigens and the complexity of immunity to TCLs derived from established tumours cannot be evaluated.

We acknowledge the reviewer's concern in our ovarian cancer model and the complexity of using TCL as the source of tumor antigens, which also resulted in difficulties in analyzing the mechanism of action. To address these concerns, we have completed a new set of experiments replacing TCL by ovalbumin as a model antigen and tested the efficacy using a B16F10-OVA melanoma model – see new Fig. 6d-n, Supplementary Fig. 25, and Supplementary Fig. 26. We demonstrate that TCL-Lip-CPMV vaccination protects from tumor challenge and importantly, we demonstrate protection from recurrence in a surgery and lung metastasis model. Further, as suggested by the reviewer, this simplified model allowed us to study the T cell response; our previous data have shown increased antigen presentation on DCs (Fig. 3h) leading to increased numbers of antigen-specific CD8+ T cells (Fig. 5, and Fig. 6g). Together with the new efficacy data, our data support the conclusion that loading antigens into liposomes followed by conjugation of CPMV creates a nanovaccine formulation that is effective to deliver antigens and adjuvants to APCs for antigen presentation and APCs activation, which is essential to produce antigen specific T cells to recognize and kill cancer cells.

The ovalbumin system illustrates that the nanoparticles work for immunity against ovalbumin but this is only preliminary data for the evaluation of antigen-specific anti-tumour immune response, similar in-depth data for which is not provided.

To address the concerns, we have completed new experiments using the B16F10-OVA melanoma model to confirm our hypothesis that OVA-Lip-CPMV can lead to effective immune protection against OVA expressing melanoma. Indeed OVA-Lip-CPMV vaccination proved efficacious leading to efficacy in preventing mice from tumor challenge and outgrowth of lung metastasis. Finally, efficacy was correlated with mechanism studies using ELISpot and flow cytometry analysis (all data is shown in updated Fig. 6d-n, Supplementary Fig. 25, and Supplementary Fig. 26).

In the clinical setting women with ovarian cancer are diagnosed with advanced stage disease, with cancer already disseminated at the time of diagnosis. Administration of the viral constructs three days prior to malignant cell injection, does not replicate the setting in human disease, a more relevant design is needed to assess the efficiency of the construct as prophylactic for recurrent disease.

We thank the reviewer for this comment. Indeed, in the clinic, women are generally diagnosed at late stages with metastatic diseases. The predominate treatment is the combination of surgical tumor removal and chemotherapy, which often leads to a short period of remission. We understand that there is no perfect mouse model to replicate the clinical. Therefore, we started our first dose of vaccination on day -3 to treat day -3 to day 0 as the remission period post-surgery. On day 0, we inoculated tumors to mimic the cancer recurrence.

To further address the concerns about the mouse models, we have included a new set of studies using the B16F10-OVA melanoma model (see updated Fig. 6d-n, Supplementary Fig. 25, and Supplementary Fig. 26) – using this model allowed to

perform surgical tumor removal and then follow recurrence by measuring lung metastatic burden (tumor rechallenge).

Although the nanoparticle system is comprehensive, the study in its current form does not provide sufficiently strong evaluation of this technology in anti-tumour immunity in ovarian cancer.

We appreciate the reviewer's comment. Our previous data (Fig. 4) showed that TCL-Lip-CPMV led to improved infiltration of adaptive immune cells into the i.p. space and improved survival benefits in our ovarian cancer model. TCL is too complex to facilitate the analysis of specific immune cells. To overcome this challenge, we added a new model to our updated manuscript using B16F10-OVA melanoma. Our new data (Fig. 6d-n) and previous data (Fig. 3h and Fig. 4) together provide a more complete picture that delivers tumor antigens and CPMV adjuvant synchronously could improve antigen processing and presentation for antigen-specific T cells production (see new data shown in updated Fig. 6 g-i).

Reviewer #3 (Remarks to the Author):

The manuscript entitled "A cowpea mosaic virus adjuvant conjugated to liposomes loaded with tumor cell lysates as an ovarian cancer vaccine" described a cancer nanovaccine composed of TCL, liposomes, and CPMV.

First of all, the novelty of this kind nanovaccine is limited as pointed by reviewer 1.

While we acknowledge that various components, plant virus, TCL and liposomes have been studied as vaccine formulations – the combination of the CPMV adjuvant with TCL-loaded liposome is novel. Through combination of the components, we synthesized a novel nanovaccine formulation which proved to be more efficacious compared to the individual components. New experiments including additional animal models highlight the potency of the approach.

Second, the challenged tumor models, such as established subcutaneous tumor models or relapse tumor models post surgery, could be used to evaluate the efficacy potency of the nanovaccine. BTW, in the current treatment model, the mice were administered by 6 doses of TCL-Lip-CPMV, which were rarely used in the study of cancer vaccines (normally, 2~3 doses).

We thank the reviewer's suggestion, and we have addressed this through new animal studies using B16F10-OVA melanoma model – the studies demonstrate that the TCL-Lip-CPMV formulation successfully protected from tumor challenge. In addition, our new studies showed that the TCL-Lip-CPMV vaccine protects from tumor recurrence (lung metastasis) post-surgery. The new data is shown in updated Fig. 6d-n.

In terms of the numbers of doses: our previously published studies using CPMV adjuvants established the requirement of 6 weekly doses to achieve efficacy against the highly aggressive ID8-Defb29/Vegf-a-Luc ovarian cancer model [1]. Thus, we followed our previous protocol for the ovarian cancer model. However, we note that for the B16F10-OVA model, only 2-3 doses were used to achieve significant benefit in preventing tumor growth, improving survival, and lung metastasis.

[1] Wang, Chao, Steven N. Fiering, and Nicole F. Steinmetz. "Cowpea mosaic virus promotes anti-tumor activity and immune memory in a mouse ovarian tumor model." *Advanced therapeutics* 2.5 (2019): 1900003.

At last, the immunology mechanism of CPMV as adjuvant and safety profile study could be strengthened. For example, adding another adjuvant (e.g. CPG or Pam 3) as control, detection of tumor specific CD8+ T cell in peripheral blood or LNs, detection of AST and ALT.

We thank the reviewer's suggestion. In this current work, our goal is to establish a new technology by combining two nanotechnology systems for antigen and adjuvant co-delivery. Using CpG or Pam 3 is out of scope for this study. In our following work, we will systematically investigate whether different adjuvants will lead to different outcomes including the tumor specific T cells in peripheral blood and LNs. Also, our histology data showed no damage to organs (Fig. 7c) and our previously published studies on CPMV have shown no effects on liver damage in terms of the AST and ALT assays [1], thus we did not carry out these studies.

[1] Chung, Young Hun, et al. "S100A9-Targeted Cowpea Mosaic Virus as a Prophylactic and Therapeutic Immunotherapy against Metastatic Breast Cancer and Melanoma." *Advanced Science* 8.21 (2021): 2101796.

Reviewer #4 (Remarks to the Author):

The author has already satisfactorily addressed my question and it can be published in its current form.

We thank the reviewer for this comment.

Dear Reviewers:

We appreciate your comments and questions. We have carefully reviewed your comments. A point-by-point response is shown below. We have revised our manuscript, and the revisions are **highlighted in yellow**.

We thank you for your time and hope you will find it acceptable for publication in Nature Communications.

Best wishes,

Nicole F. Steinmetz PhD
Professor, UC San Diego

REVIEWER COMMENTS

Reviewer #1 (Remarks to the Author):

The authors have completed additional experiments using the B16.OVA tumour model to look at antigen-specific tumour responses. This strengthens the manuscript and brings it in line with other similar scientific papers reporting on cancer/vaccines.

We thank the reviewer for this comment.

While looking at a model tumour antigen allows for clear examination of antigen-specific responses and prophylactic treatment experiments. Perhaps a better way to complete these studies would have to use tumour lysate from the B16 cell line and assay e.g. with ELISPOT, several of the endogenous tumour antigens which have been reported in the literature for the B16 cells. As these often fall somewhat short of the results obtainable using OVA and this method would be more in line with the authors previous strategy of using TCL.

We appreciate the reviewer's suggestion. The ELIspot experiment demonstrates that OVA-Lip-CPMV vaccination primes the establishment of a tumor antigen (OVA)-specific immunity, therefore preventing cancer onset and recurrence (as shown in the new data presented in the previous revision). We agree that the study of tumor lysate would add value to the project, and we will consider this in future work (in line with the editor's comment that additional animal models were not requested at this stage). To further augment support of efficacy of the nanovaccine, we carried out additional ELIspot assays to confirm that splenocytes recognize live B16F10-OVA cells (and not just the OVA antigen as was previously shown). This data is shown in the **updated manuscript (line 470-473) and Supplementary Fig. 26d**. The results are consistent with the flow cytometry experiments to examine the IFN- \$\gamma\$, and Granzyme B expression within CD8 T cells after incubating splenocytes with the B16F10-OVA cells (Fig. 6i,j). The new data is in line with the hypothesis and clearly shows that splenocytes from the OVA-Lip-CPMV

group show stronger immune response against B16F10-OVA cells compared to control demonstrated by the overwhelming IFN- γ red spots and area.

The use of orthotopic (i.d.) melanoma engraftment has been reported to lead to metastasis to local lymph nodes and other organs once tumours become large. The surgical model employed in the revisions is somewhat unusual as it involves surgical removal of a small primary (d7), most likely prior to any met formation, followed by vaccination and then artificial seeding of lung mets using the i.v. inoculation route (A common prophylactic approach). In this scenario, I wonder why the earlier surgery needs to be performed and what effect the authors think it might be having on the vaccination. Presumably there is some endogenous response to the first tumour that the vaccination can boost? Where they setting the effect of prior surgery on the vaccination? However, this is not really representative of surgical resection with residual primary or metastatic disease, developing alongside the primary tumour.

A surgery ovarian tumor model when tumors are disseminated to the IP space is technically not feasible – our goal was to establish a model that mimics treatment of ovarian cancer. Therefore, we established B16F10-OVA tumors, removed the tumors by surgery – to ensure establishment of metastasis we challenged the mice with the same tumor cells IV. In the clinic ovarian cancer patients undergo surgery usually followed with chemotherapy – patients are then in a short period of remission, but eventually tumors recurrence occurs. Our model, which is like all models not perfect, was set to mimic this clinical scenario – vaccination during remission. Of note, while literature reports demonstrate that the B16F10 model leads to lungs mets, in our model – even when tumors are grown to endpoint – the establishment of lung tumors is not highly reliable leading to high variation and statistically challenging to analyze data, this is why cells were administered IV. The overall goal of the proposed nanovaccine is to prevent recurrence and clinically, we propose treatment in the remission phase to prevent recurrence and metastatic disease. Now we have emphasized our rationale as highlighted in yellow in our revised manuscript (line 26-35, line 62-72, line 395-403, line 550-555).

Reviewer #2 (Remarks to the Author):

The authors have conducted additional work that has improved several parts of the manuscript and demonstrated vaccine efficacy in a surgical model of B16F10-ovalbumin expressing melanoma tumours. However, my concern remains on the data supporting the efficacy of TCL_Lip_CPMV complexes in ovarian cancer as the manuscript still relies solely on the ID8-Defb29/Vegf-a-Luc model. This model has several critical limitations as an ovarian cancer model, including the expression of the weakly immunogenic firefly luciferase which, in the context of CPMV, could be driving the observed outcomes.

We acknowledge the reviewer's concern. We understand that there is no perfect murine ovarian tumor model to represent human diseases. However, the ID8-Defb29/Vegf-a-Luc closely resembles some characteristics of human ovarian cancer, such as ascites and peritoneal metastasis. We understand the shortcomings of any model pursued and are currently establishing additional ovarian tumor models in the lab – these will be subject to future investigation. With regards to the concerns about the weakly immunogenic firefly luciferase in the ID8-Defb29/Vegf-a-Luc model and use of CPMV, these concerns were alleviated by demonstration of efficacy in the B16F10 model – the nanovaccine was administered s.c. and therefore CPMV's previously described adjuvant efficacy by remodeling the tumor microenvironment is ruled out. Rather, in the present study we focused CPMV's adjuvant properties to stimulate immune responses against co-delivered antigens and collectively our data support efficacy of this approach in ovarian and melanoma tumor models.

Reviewer #3 (Remarks to the Author):

I continue to harbor reservations regarding the efficacy of the nanovaccine. Specifically, the experimental design appears unconventional, as it involves administering a single injection prior to tumor inoculation. In clinical practice, it is impractical to develop personalized cancer vaccines before tumors are detected.

We acknowledge the reviewer's comment and agree there is no perfect tumor mouse model. A surgery ovarian tumor model when tumors are disseminated to the IP space is technically not feasible – our goal was to establish a model that mimics treatment of ovarian cancer. Therefore, we established B16F10-OVA tumors, removed the tumors by surgery – to ensure establishment of metastasis we challenged the mice with the same tumor cells IV. In the clinic ovarian cancer patients undergo surgery usually followed with chemotherapy – patients are then in a short period of remission, but eventually tumors recurrence occurs. Our model, which is like all models not perfect, was set to mimic this clinical scenario – vaccination during remission. Of note, while literature reports demonstrate that the B16F10 model leads to lungs mets, in our model – even when tumors are grown to endpoint – the establishment of lung tumors is not highly reliable leading to high variation and statistically challenging to analyze data, this is why cells were administered IV. The overall goal of the proposed nanovaccine is to prevent recurrence and clinically, we propose treatment in the remission phase to prevent recurrence and metastatic disease. Now we have emphasized our rationale as highlighted in yellow in our revised manuscript (line 26-35, line 62-72, line 395-403, line 550-555).

Despite this unusual setup, the statistical analysis revealed no significant differences between the TCL-Lip + CPMV group and the TCL-Lip-CPMV or CPMV-only groups ($p > 0.05$), as illustrated in Figures 6a and 6b.

Although TCL-Lip-CPMV did not show significant survival benefits compared to CPMV and TCL-Lip + CPMV, it did show a trend in preventing tumor onset with 5 out of 8 mice free of tumors. CPMV alone as an intratumoral agent has exceptional efficacy as it remodels the tumor microenvironment priming anti-tumor responses. These concerns about the overpowering efficacy of the CPMV adjuvant are alleviated by testing in the B16F10 tumor model – here, the nanovaccine was administered s.c. and therefore CPMV's previously described adjuvant efficacy by remodeling the tumor microenvironment is ruled out. Rather, the B16F10 study demonstrated CPMV's adjuvant properties to stimulate immune responses against co-delivered antigens and collectively our data support efficacy of this approach in ovarian and melanoma tumor models.

Similarly, the new B16F10-OVA model failed to provide novel insights into the vaccine's efficacy, particularly in the context of ovarian cancer.

While the B16F10-OVA is not an ovarian cancer model – our efforts were to set up a model that would mimic the clinical treatment scheme of ovarian cancer: surgery to get to a remission phase – during which treatment was administered – to assay whether the nanovaccine would prevent recurrence (here modeled by IV challenge of the same tumor cells). We designed this model, due to the limitations and challenges of survival surgery of IP the disseminated ovarian tumor model. The choice of the B16F10-OVA model allowed us to utilize OVA to gain additional insights into the mechanism of action.

Finally, while the treatment model depicted in Figures 6k-n did demonstrate some potential efficacy compared to the control, the appropriate control group should have been TCL-Lip-CPMV or CPMV-only, rather than HEPES.

We acknowledge the reviewer's comment. Data consistently showed enhanced efficacy of the TCL-Lip-CPMV and OVA-Lip-CPMV formulation vs control groups. To minimize the number of animals and costs it was decided to only test OVA-Lip-CPMV to answer whether this candidate formulation could prevent onset of mets after vaccination. The data support this – unfortunately without further animal studies, comparisons to other groups cannot be made. We have added clarification and highlight the shortcomings of this study in the revised manuscript (line 480-487, line 491-493).

In summary, the authors have not provided sufficient evidence to substantiate the efficacy of TCL-Lip-CPMV as an ovarian cancer vaccine.

We appreciate the concerns about the lack of better ovarian tumor models – our lab is now making progress to establish novel ovarian tumor models and we will report on in

future work under a separate cover. Because of this we turned toward the B16F10-OVA model – while not ovarian-cancer specific, this model allowed us to ask important question and provides further evidence on the efficacy of the nanovaccine and its mechanism of action. We added further discussion in the *Conclusions section* (line 550-555).

Dear Reviewers:

We appreciate your comments and questions. We have carefully reviewed your comments. A point-by-point response is shown below. We have added additional information in our manuscript that is highlighted in yellow.

We thank you for your time and hope you will find it acceptable for publication in Nature Communications.

Best wishes,

Nicole F. Steinmetz PhD
Professor, UC San Diego

REVIEWER COMMENTS

Reviewer #1 (Remarks to the Author):

Regarding the authors response to my comments (R1):

I appreciate the updated rationale for the proposed nanovaccine as this helps to clarify the intended translational use case without downplaying the limitations of the study and models used throughout.

We thank the reviewer for this comment.

Considering the ELISPOT experiment, it would have been interesting to examine the response against both the B16.OVA-expressing cell line and parental B16 cell line in co-culture experiments to determine the extent (if any) of epitope spreading that is induced by the nanovaccine. I appreciate that these experiments used vaccinated mice as the source of splenocytes, making additional experiments difficult without using more animals. This data would be of interest in the context of cancer vaccination in patients, where dominant antigens are often the target of downregulation by tumours under immune pressure and critical e.g. in relapsed/metastasised cases.

We acknowledge the reviewer's comment and suggestion and agree that this would be a useful additional experiments. At the present time however we do not have tissues to conduct this study – thus we must defer to conduct these experiments under a different cover and will test whether our liposome and CPMV conjugated nanovaccine could lead to antigen expansion. We have added this additional information in our revised manuscript (line 476-482). Regarding the potential downregulated tumor antigens in relapsed diseases, we could use “immunological cold” mouse models in our future work to assess whether our nanovaccine could effectively prevent tumor onset and recurrence.

Additionally, for Supp. Figure 26, why is there no OVA-Lip group in (c/d) and no OVA-lip-CPMV group in (e-g) for comparison. I apologise if I am missing something here.

Throughout the study, we observed the OVA-Lip-CPMV performed the best with 6 out of 10 mice rejecting B16F10-OVA tumor growth in both the challenge and rechallenge studies, whereas only 1 out of 10 mice from the OVA-Lip group survived. Statistically, OVA-Lip vaccination failed to achieve protection against tumor challenge similar to the HEPES control group. It is not possible to perform statistical analysis and comparison between the OVA-Lip (1 mouse) and OVA-Lip-CPMV (6 mice) groups due to the differences in group sizes (with one group only having 1 mouse). Therefore, we decided to graph them differently and focus our story mainly on the OVA-Lip-CPMV group.

Considering the surgical model of relapse/remission; in the clinical setting, can the authors confirm whether recurrence is predominantly local or via cryptic metastatic disease. The later case is modelled in their experimental design but not the setting of local relapse, which while difficult to test in the ovarian setting, could certainly have been modelled in the B16 flank model or common breast cancer models.

Based on literatures [1,2], ovarian cancer recurrence mostly occurs in the peritoneal cavity, but it can also metastasize to pelvis, lymph nodes, lungs, etc. We thank the reviewer for the suggestion to use other models to test local relapses. We will test out the suggested models in our future work.

[1] Dao, Minh D., et al. "Recurrence patterns after extended treatment with bevacizumab for ovarian, fallopian tube, and primary peritoneal cancers." *Gynecologic oncology* 130.2 (2013): 295-299.

[2] Amate, Pascale, et al. "Ovarian cancer: sites of recurrence." *International Journal of Gynecological Cancer* 23.9 (2013): 1590-1596.

In either scenario human patients will not be truly disease free during remission, as the mice are in this case, and this is likely to be important for vaccination i.e. via ongoing cancer-induced immunomodulation and as a source of additional antigen.

We agree with the reviewer. In the clinical setting, after surgical removal of most of the tumor tissue, patients still need chemotherapy to treat leftover and non-visible tumors. In our future work, we will carry out studies as suggested by the reviewer by establishing the tumor first followed by our nanovaccine treatment.

Reviewer #3 (Remarks to the Author):

While the authors have acknowledged certain limitations in the data quality regarding efficacy of the nanovaccine and offered explanations that may appear somewhat strained, they have not taken concrete steps to address these scientific concerns. On this basis, I remain unconvinced that sufficient evidence has been provided to substantiate the efficacy of TCL-Lip-CPMV as a viable ovarian cancer vaccine.

We appreciate the reviewer's critique and comment. We will continue optimizing our liposome and CPMV conjugated nanovaccine, carrying out well-designed experiments, and improving its efficacy using additional mouse models in our future work.